# CATs: Cost Aggregation Transformers for Visual Correspondence

**Seokju Cho**[*]
Yonsei University

**Sunghwan Hong**[*]
Korea University

**Sangryul Jeon**
Yonsei University

**Yunsung Lee**
Korea University

**Kwanghoon Sohn**
Yonsei University

**Seungryong Kim**[†]
Korea University

## Abstract

We propose a novel cost aggregation network, called Cost Aggregation Transformers (CATs), to find dense correspondences between semantically similar images with additional challenges posed by large intra-class appearance and geometric variations. Cost aggregation is a highly important process in matching tasks, which the matching accuracy depends on the quality of its output. Compared to hand-crafted or CNN-based methods addressing the cost aggregation, in that either lacks robustness to severe deformations or inherit the limitation of CNNs that fail to discriminate incorrect matches due to limited receptive fields, CATs explore global consensus among initial correlation map with the help of some architectural designs that allow us to fully leverage self-attention mechanism. Specifically, we include appearance affinity modeling to aid the cost aggregation process in order to disambiguate the noisy initial correlation maps and propose multi-level aggregation to efficiently capture different semantics from hierarchical feature representations. We then combine with swapping self-attention technique and residual connections not only to enforce consistent matching, but also to ease the learning process, which we find that these result in an apparent performance boost. We conduct experiments to demonstrate the effectiveness of the proposed model over the latest methods and provide extensive ablation studies. Code and trained models are available at `https://sunghwanhong.github.io/CATs/`.

## 1 Introduction

Establishing dense correspondences across semantically similar images can facilitate many Computer Vision applications, including semantic segmentation [46, 54, 36], object detection [29], and image editing [53, 30, 28, 25]. Unlike classical dense correspondence problems that consider visually similar images taken under the geometrically constrained settings [16, 19, 50, 18], semantic correspondence poses additional challenges from large intra-class appearance and geometric variations caused by the unconstrained settings of given image pair.

Recent approaches [42, 43, 45, 34, 37, 39, 31, 58, 47, 57, 51, 35] addressed these challenges by carefully designing deep convolutional neural networks (CNNs)-based models analogously to the classical matching pipeline [48, 41], feature extraction, cost aggregation, and flow estimation. Several works [24, 9, 37, 39, 47, 51] focused on the feature extraction stage, as it has been proven that the more powerful feature representation the model learns, the more robust matching is obtained [24, 9, 51]. However, solely relying on the matching similarity between features without any prior often suffers

---

[*]Equal contribution
[†]Corresponding author

from the challenges due to ambiguities generated by repetitive patterns or background clutters [42, 24, 26]. On the other hand, some methods [42, 49, 43, 23, 26, 58] focused on flow estimation stage either by designing additional CNN as an ad-hoc regressor that predicts the parameters of a single global transformation [42, 43], finding confident matches from correlation maps [20, 26], or directly feeding the correlation maps into the decoder to infer dense correspondences [58]. However, these methods highly rely on the quality of the initial correlation maps.

The latest methods [45, 37, 44, 21, 31, 27, 35] have focused on the second stage, highlighting the importance of cost aggregation. Since the quality of correlation maps is of prime importance, they proposed to refine the matching scores by formulating the task as optimal transport problem [47, 31], re-weighting matching scores by Hough space voting for geometric consistency [37, 39], or utilizing high-dimensional 4D or 6D convolutions to find locally consistent matches [45, 44, 27, 35]. Although formulated variously, these methods either use hand-crafted techniques that are neither learnable nor robust to severe deformations, or inherit the limitation of CNNs, e.g., limited receptive fields, failing to discriminate incorrect matches that are locally consistent.

In this work, we focus on the cost aggregation stage, and propose a novel cost aggregation network to tackle aforementioned issues. Our network, called Cost Aggregation with Transformers (CATs), is based on Transformer [61, 10], which is renowned for its global receptive field. By considering all the matching scores computed between features of input images globally, our aggregation networks explore global consensus and thus refine the ambiguous or noisy matching scores effectively.

Specifically, based on the observation that desired correspondence should be aligned at discontinuities with appearance of images, we concatenate an appearance embedding with the correlation map, which helps to disambiguate the correlation map within the Transformer. To benefit from hierarchical feature representations, following [26, 39, 58], we use a stack of correlation maps constructed from multi-level features, and propose to effectively aggregate the scores across the multi-level correlation maps. Furthermore, we consider bidirectional nature of correlation map, and leverage the correlation map from both directions, obtaining reciprocal scores by swapping the pair of dimensions of correlation map in order to allow global consensus in both perspective. In addition to all these combined, we provide residual connections around aggregation networks in order to ease the learning process.

We demonstrate our method on several benchmarks [38, 11, 12]. Experimental results on various benchmarks prove the effectiveness of the proposed model over the latest methods for semantic correspondence. We also provide an extensive ablation study to validate and analyze components in CATs.

## 2   Related Work

**Semantic Correspondence.**   Methods for semantic correspondence generally follow the classical matching pipeline [48, 41], including feature extraction, cost aggregation, and flow estimation. Most early efforts [7, 30, 11] leveraged the hand-crafted features which are inherently limited in capturing high-level semantics. Though using deep CNN-based features [5, 24, 42, 43, 23, 49, 26] has become increasingly popular thanks to their invariance to deformations, without a means to refine the matching scores independently computed between the features, the performance would be rather limited.

To alleviate this, several methods focused on flow estimation stage. Rocco et al. [42, 43] proposed an end-to-end network to predict global transformation parameters from the matching scores, and their success inspired many variants [49, 23, 25]. RTNs [23] obtain semantic correspondences through an iterative process of estimating spatial transformations. DGC-Net [34], Semantic-GLU-Net [58] and DMP [15] utilize a CNN-based decoder to directly find correspondence fields. PDC-Net [59] proposed a flexible probabilistic model that jointly learns the flow estimation and its uncertainty. Arguably, directly regressing correspondences from the initial matching scores highly relies on the quality of them.

Recent numerous methods [45, 37, 39, 31, 47, 51, 35] thus have focused on cost aggregation stage to refine the initial matching scores. Among hand-crafted methods, SCOT [31] formulates semantic correspondence as an optimal transport problem and attempts to solve two issues, namely many to one matching and background matching. HPF [37] first computes appearance matching confidence using hyperpixel features and then uses Regularized Hough Matching (RHM) algorithm for cost aggregation to enforce geometric consistency. DHPF [39], that replaces feature selection algorithm

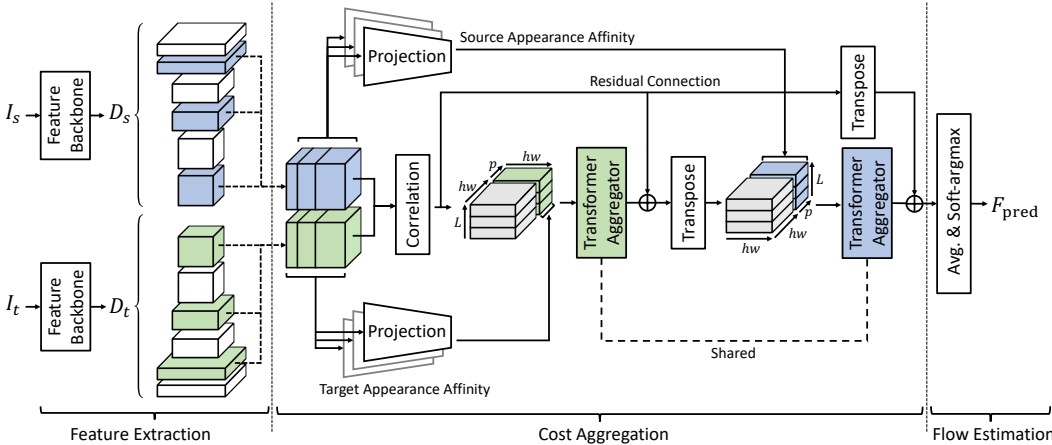

Figure 1: **Overall network architecture.** Our networks consist of feature extraction, cost aggregation, and flow estimation modules. We first extract multi-level dense features and construct a stack of correlation maps. We then concatenate with embedded features and feed into the Transformer-based cost aggregator to obtain a refined correlation map. The flow is then inferred from the refined map.

of HPF [37] with trainable networks, also uses RHM. However, these hand-crafted techniques for refining the matching scores are neither learnable nor robust to severe deformations. As learning-based approaches, NC-Net [45] utilizes 4D convolution to achieve local neighborhood consensus by finding locally consistent matches, and its variants [44, 27] proposed more efficient methods. GOCor [57] proposed aggregation module that directly improves the correlation maps. GSF [21] formulated pruning module to suppress false positives of correspondences in order to refine the initial correlation maps. CHM [35] goes one step further, proposing a learnable geometric matching algorithm which utilizes 6D convolution. However, they are all limited in the sense that they inherit limitation of CNN-based architectures, which is local receptive fields.

**Transformers in Vision.** Transformer [61], the *de facto* standard for Natural Language Processing (NLP) tasks, has recently imposed significant impact on various tasks in Computer Vision fields such as image classification [10, 55], object detection [3, 62], tracking and matching [52, 51]. ViT [10], the first work to propose an end-to-end Transformer-based architecture for the image classification task, successfully extended the receptive field, owing to its self-attention nature that can capture global relationship between features. For visual correspondence, LoFTR [51] uses cross and self-attention module to refine the feature maps conditioned on both input images, and formulate the hand-crafted aggregation layer with dual-softmax [45, 60] and optimal transport [47] to infer correspondences. COTR [22] takes coordinates as an input and addresses dense correspondence task without the use of correlation map. Unlike these, for the first time, we propose a Transformer-based cost aggregation module.

## 3 Methodology

### 3.1 Motivation and Overview

Let us denote a pair of images, i.e., source and target, as $I_s$ and $I_t$, which represent semantically similar images, and features extracted from $I_s$ and $I_t$ as $D_s$ and $D_t$, respectively. Here, our goal is to establish a dense correspondence field $F(i)$ between two images that is defined for each pixel $i$, which warps $I_t$ towards $I_s$.

Estimating the correspondence with sole reliance on matching similarities between $D_s$ and $D_t$ is often challenged by the ambiguous matches due to the repetitive patterns or background clutters [42, 24, 26]. To address this, numerous methods proposed cost aggregation techniques that focus on refining the initial matching similarities either by formulating the task as optimal transport problem [47, 31], using regularized Hough matching to re-weight the costs [37, 39], or 4D or 6D convolutions [45, 27, 44, 35]. However, these methods either use hand-crafted techniques that are weak to severe deformations, or fail to discriminate incorrect matches due to limited receptive fields.

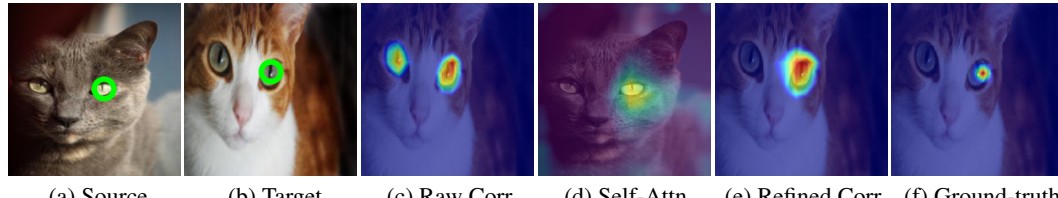

| (a) Source | (b) Target | (c) Raw Corr. | (d) Self-Attn. | (e) Refined Corr. | (f) Ground-truth |

Figure 2: **Visualization of correlation map and self-attention:** (a) source image, (b) target image, (c) raw correlation map, (d) self-attention, (e) refined correlation map, and (f) ground-truth, which are bilinearly upsampled. The visualization proves that CATs successfully aggregates the costs by integrating the surrounding information of the query, represented as green circle in the source.

To overcome these, we present Transformer-based cost aggregation networks that effectively integrate information present in all pairwise matching costs, dubbed CATs, as illustrated in Fig. 1. As done widely in other works [42, 45, 50, 34, 37], we follow the common practice for feature extraction and cost computation. In the following, we first explain feature extraction and cost computation, and then describe several critical design choices we made for effective aggregation of the matching costs.

### 3.2 Feature Extraction and Cost Computation

To extract dense feature maps from images, we follow [26, 37, 39] that use multi-level features for construction of correlation maps. We use CNNs that produce a sequence of $L$ feature maps, and $D^l$ represents a feature map at $l$-th level. As done in [37], we use different combination of multi-level features depending on the dataset trained on, e.g., PF-PASCAL [12] or SPair-71k [38]. Given a sequence of feature maps, we resize all the selected feature maps to $\mathbb{R}^{h \times w \times c}$, with height $h$, width $w$, and $c$ channels. The resized features then undergo $l$-2 normalization.

Given resized dense features $D_s$ and $D_t$, we compute a correlation map $\mathcal{C} \in \mathbb{R}^{hw \times hw}$ using the inner product between features: $\mathcal{C}(i, j) = D_t(i) \cdot D_s(j)$ with points $i$ and $j$ in the target and source features, respectively. In this way, all pairwise feature matches are computed and stored. However, raw matching scores contain numerous ambiguous matching points as exemplified in Fig. 2, which results inaccurate correspondences. To remedy this, we propose cost aggregation networks in the following that aim to refine the ambiguous or noisy matching scores.

### 3.3 Transformer Aggregator

Renowned for its global receptive fields, one of the key elements of Transformer [61] is the self-attention mechanism, which enables finding the correlated input tokens by first feeding into scaled dot product attention function, normalizing with Layer Normalization (LN) [1], and passing the normalized values to a MLP. Several works [10, 3, 62, 51] have shown that given images or features as input, Transformers [61] integrate the global information in a flexible manner by learning to find the attention scores for all pairs of tokens.

In this paper, we leverage the Transformers to integrate the matching scores to discover global consensus by considering global context information. Specifically, we obtain a refined cost $\mathcal{C}'$ by feeding the raw cost $\mathcal{C}$ to the Transformer $\mathcal{T}$, consisting of self-attention, LN, and MLP modules:

$$\mathcal{C}' = \mathcal{T}(\mathcal{C} + E_{\text{pos}}), \tag{1}$$

where $E_{\text{pos}}$ denotes positional embedding. The standard Transformer receives as input a 1D sequence of token embeddings. In our context, we reshape the correlation map $\mathcal{C}$ into a sequence of vectors $\mathcal{C}(k) \in \mathbb{R}^{1 \times hw}$ for $k \in \{1, ..., hw\}$. We visualize the refined correlation map with self-attention in Fig. 2, where the ambiguities are significantly resolved.

**Appearance Affinity Modeling.** When only matching costs are considered for aggregation, self-attention layer processes the correlation map itself disregarding the noise involved in the correlation map, which may lead to inaccurate correspondences. Rather than solely relying on raw correlation map, we additionally provide an appearance embedding from input features to disambiguate the correlation map aided by appearance affinity within the Transformer. Intuition behind is that visually similar points in an image, e.g., color or feature, have similar correspondences, as proven in stereo matching literature, e.g., Cost Volume Filtering (CVF) [16, 50].

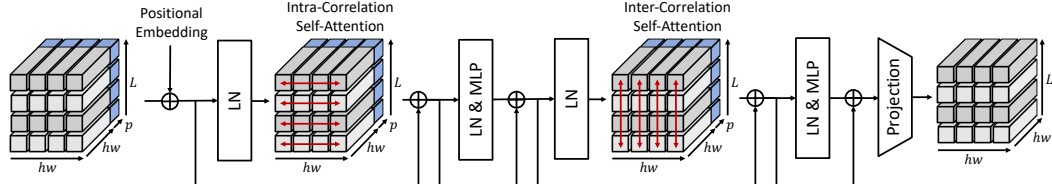

Figure 3: **Illustration of Transformer aggregator.** Given correlation maps $\mathcal{C}$ with projected features, Transformer aggregation consisting of intra- and inter-correlation self-attention with LN and MLP refines the inputs not only across spatial domains but across levels.

To provide appearance affinity, we propose to concatenate embedded features projected from input features with the correlation map. We first feed the features $D$ into linear projection networks, and then concatenate the output along corresponding dimension, so that the correlation map is augmented such that $[\mathcal{C}, \mathcal{P}(D)] \in \mathbb{R}^{hw \times (hw+p)}$, where $[\,\cdot\,]$ denotes concatenation, $\mathcal{P}$ denotes linear projection networks, and $p$ is channel dimension of embedded feature. Within the Transformer, self-attention layer aggregates the correlation map and passes the output to the linear projection networks to retain the size of original correlation $\mathcal{C}$.

**Multi-Level Aggregation.** As shown in [37, 34, 39, 58, 31], leveraging multi-level features allows capturing hierarchical semantic feature representations. Thus we also use multi-level features from different levels of convolutional layers to construct a stack of correlation maps. Each correlation map $\mathcal{C}^l$ computed between $D_s^l$ and $D_t^l$ is concatenated with corresponding embedded features and fed into the aggregation networks. The aggregation networks now consider multiple correlations, aiming to effectively aggregates the matches by the hierarchical semantic representations.

As shown in Fig. 3, a stack of $L$ augmented correlation maps, $[\mathcal{C}^l, \mathcal{P}(D^l)]_{l=1}^L \in \mathbb{R}^{hw \times (hw+p) \times L}$, undergo the Transformer aggregator. For each $l$-th augmented correlation map, we aggregate with self-attention layer across all the points in the augmented correlation map, and we refer this as *intra*-correlation self-attention. In addition, subsequent to this, the correlation map undergoes *inter*-correlation self-attention across multi-level dimensions. Contrary to HPF [37] that concatenates all the multi-level features and compute a correlation map, which disregards the level-wise similarities, within the inter-correlation layer of the proposed model, the similar matching scores are explored across multi-level dimensions. In this way, we can embrace richer semantics in different levels of feature maps, as shown in Fig. 4.

### 3.4 Cost Aggregation with Transformers

By leveraging the Transformer aggregator, we present cost aggregation framework with following additional techniques to improve the performance.

**Swapping Self-Attention.** To obtain a refined correlation map invariant to order of the input images and impose consistent matching scores, we argue that reciprocal scores should be used as aids to infer confident correspondences. As correlation map contains bidirectional matching scores, from both target and source perspective, we can leverage matching similarities from both directions in order to obtain more reciprocal scores as done similarly in other works [45, 26].

As shown in Fig. 1, we first feed the augmented correlation map to the aforementioned Transformer aggregator. Then we transpose the output, swapping the pair of dimensions in order to concatenate with the embedded feature from the other image, and feed into the subsequent another aggregator. Note that we share the parameters of the Transformer aggregators to obtain reciprocal scores. Formally, we define the whole process as following:

$$
\begin{aligned}
\mathcal{S} &= \mathcal{T}([\mathcal{C}^l, \mathcal{P}(D_t^l)]_{l=1}^L + E_{\text{pos}}), \\
\mathcal{C}' &= \mathcal{T}([(\mathcal{S}^l)^{\text{T}}, \mathcal{P}(D_s^l)]_{l=1}^L + E_{\text{pos}}),
\end{aligned}
\tag{2}
$$

where $\mathcal{C}^{\text{T}}(i,j) = \mathcal{C}(j,i)$ denotes swapping the pair of dimensions corresponding to the source and target images; $\mathcal{S}$ denotes the intermediate correlation map before swapping the axis. Note that NC-Net [45] proposed a similar procedure, but instead of processing serially, they separately process the correlation map and its transposed version and add the outputs, which is designed to produce

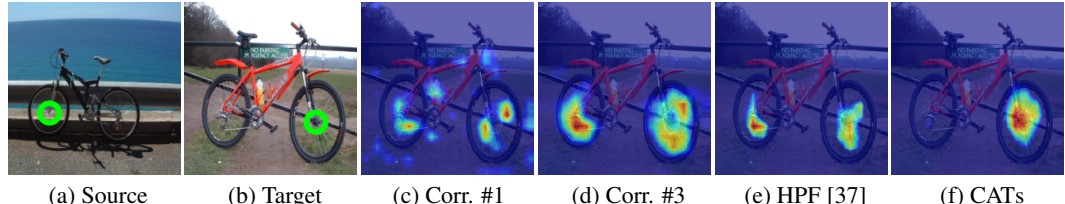

| (a) Source | (b) Target | (c) Corr. #1 | (d) Corr. #3 | (e) HPF [37] | (f) CATs |

Figure 4: **Visualization of multi-level aggregation:** (a) source, (b) target images, (c), (d) multi-level correlation maps (e.g., $l = 1$ and $l = 3$), respectively, and final correlation maps by (e) HPF [37] and (f) CATs. Note that HPF and CATs utilize the same feature maps. Compared to HPF, CATs successfully embrace richer semantics in different levels of feature map.

a correlation map invariant to the particular order of the input images. Unlike this, we process the correlation map serially, first aggregating one pair of dimensions and then further aggregating with respect to the other pair. In this way, the subsequent attention layer is given more consistent matching scores as an input, allowing further reduction of inconsistent matching scores. We include an ablation study to justify our choice in Section 4.4

**Residual Connection.** At the initial phase when the correlation map is fed into the Transformers, noisy score maps are inferred due to randomly-initialized parameters, which could complicate the learning process. To stabilize the learning process and provide a better initialization for the matching, we employ the residual connection. Specifically, we enforce the cost aggregation networks to estimate the residual correlation by adding residual connection around aggregation networks.

### 3.5 Training

**Data Augmentation.** Transformer is well known for lacking some of inductive bias and its data-hungry nature thus necessitates a large quantity of training data to be fed [61, 10]. Recent methods [55, 56, 32] that employ the Transformer to address Computer Vision tasks have empirically shown that data augmentation techniques have positive impact on performance. However, in correspondence task, the question of to what extent can data augmentation affect the performance has not yet been properly addressed. From the experiments, we empirically find that data augmentation has positive impacts on performance in semantic correspondence with Transformers as reported in Section 4.4. To apply data augmentation [6, 2] with predetermined probabilities to input images at random. Specifically, 50% of the time, we randomly crop the input image, and independently for each augmentation function used in [6], we set the probability for applying the augmentation as 20%. More details can be found in supplementary material.

**Training Objective.** As in [37, 39, 35], we assume that the ground-truth keypoints are given for each pair of images. We first average the stack of refined correlation maps $\mathcal{C}' \in \mathbb{R}^{hw \times hw \times L}$ to obtain $\mathcal{C}'' \in \mathbb{R}^{hw \times hw}$ and then transform it into a dense flow field $F_{\mathrm{pred}}$ using soft-argmax operator [26]. Subsequently, we compare the predicted dense flow field with the ground-truth flow field $F_{\mathrm{GT}}$ obtained by following the protocol of [37] using input keypoints. For the training objective, we utilize Average End-Point Error (AEPE) [34], computed by averaging the Euclidean distance between the ground-truth and estimated flow. We thus formulate the objective function as $\mathcal{L} = \|F_{\mathrm{GT}} - F_{\mathrm{pred}}\|_2$.

## 4 Experiments

### 4.1 Implementation Details

For backbone feature extractor, we use ResNet-101 [14] pre-trained on ImageNet [8], and following [37], extract the features from the best subset layers. Other backbone features can also be used, which we analyze the effect of various backbone features in the following ablation study. For the hyper-parameters for Transformer encoder, we set the depth as 1 and the number of heads as 6. We resize the spatial size of the input image pairs to 256×256 and a sequence of selected features are resized to 16×16. We use a learnable positional embedding [10], instead of fixed [61]. We implemented our network using PyTorch [40], and AdamW [33] optimizer with an initial learning

Table 1: **Quantitative evaluation on standard benchmarks [38, 11, 12].** Higher PCK is better. The best results are in bold, and the second best results are underlined. CATs† means CATs without fine-tuning feature backbone. *Feat.-level: Feature-level, FT. feat.: Fine-tune feature.*

| Methods | Feat.-level | FT. feat. | Aggregation | SPair-71k [38] PCK @ $\alpha_{\text{bbox}}$ 0.1 | PF-PASCAL [12] PCK @ $\alpha_{\text{img}}$ | | | PF-WILLOW [11] PCK @ $\alpha_{\text{bbox}}$ | | |
| --- | --- | --- | --- | --- | --- | --- | --- | --- | --- | --- |
| | | | | | 0.05 | 0.1 | 0.15 | 0.05 | 0.1 | 0.15 |
| WTA | Single | ✗ | - | 25.7 | 35.2 | 53.3 | 62.8 | 24.7 | 46.9 | 59.0 |
| CNNGeo [42] | Single | ✗ | - | 20.6 | 41.0 | 69.5 | 80.4 | 36.9 | 69.2 | 77.8 |
| A2Net [49] | Single | ✗ | - | 22.3 | 42.8 | 70.8 | 83.3 | 36.3 | 68.8 | 84.4 |
| WeakAlign [43] | Single | ✗ | - | 20.9 | 49.0 | 74.8 | 84.0 | 37.0 | 70.2 | 79.9 |
| RTNs [23] | Single | ✗ | - | 25.7 | 55.2 | 75.9 | 85.2 | 41.3 | 71.9 | 86.2 |
| SFNet [26] | Multi | ✗ | - | - | 53.6 | 81.9 | 90.6 | 46.3 | 74.0 | 84.2 |
| NC-Net [45] | Single | ✓ | 4D Conv. | 20.1 | 54.3 | 78.9 | 86.0 | 33.8 | 67.0 | 83.7 |
| DCC-Net [17] | Single | ✗ | 4D Conv. | - | 55.6 | 82.3 | 90.5 | 43.6 | 73.8 | 86.5 |
| HPF [37] | Multi | - | RHM | 28.2 | 60.1 | 84.8 | 92.7 | 45.9 | 74.4 | 85.6 |
| GSF [21] | Multi | ✗ | 2D Conv. | 36.1 | 65.6 | 87.8 | 95.9 | 49.1 | 78.7 | _90.2_ |
| ANC-Net [27] | Single | ✗ | 4D Conv. | - | - | 86.1 | - | - | - | - |
| DHPF [39] | Multi | ✗ | RHM | 37.3 | _75.7_ | 90.7 | _95.0_ | 49.5 | 77.6 | 89.1 |
| SCOT [31] | Multi | - | OT-RHM | 35.6 | 63.1 | 85.4 | 92.7 | 47.8 | 76.0 | 87.1 |
| CHM [35] | Single | ✓ | 6D Conv. | _46.3_ | **80.1** | _91.6_ | 94.9 | **52.7** | **79.4** | 87.5 |
| CATs† | Multi | ✗ | Transformer | 42.4 | 67.5 | 89.1 | 94.9 | 46.6 | 75.6 | 87.5 |
| CATs | Multi | ✓ | Transformer | **49.9** | 75.4 | **92.6** | **96.4** | _50.3_ | _79.2_ | **90.3** |

Table 2: **Per-class quantitative evaluation on SPair-71k [38] benchmark.**

| Methods | aero. | bike | bird | boat | bott. | bus | car | cat | chai. | cow | dog | hors. | mbik. | pers. | plan. | shee. | trai. | tv | all |
| --- | --- | --- | --- | --- | --- | --- | --- | --- | --- | --- | --- | --- | --- | --- | --- | --- | --- | --- | --- |
| CNNGeo [42] | 23.4 | 16.7 | 40.2 | 14.3 | 36.4 | 27.7 | 26.0 | 32.7 | 12.7 | 27.4 | 22.8 | 13.7 | 20.9 | 21.0 | 17.5 | 10.2 | 30.8 | 34.1 | 20.6 |
| A2Net [49] | 22.6 | 18.5 | 42.0 | 16.4 | 37.9 | 30.8 | 26.5 | 35.6 | 13.3 | 29.6 | 24.3 | 16.0 | 21.6 | 22.8 | 20.5 | 13.5 | 31.4 | 36.5 | 22.3 |
| WeakAlign [43] | 22.2 | 17.6 | 41.9 | 15.1 | 38.1 | 27.4 | 27.2 | 31.8 | 12.8 | 26.8 | 22.6 | 14.2 | 20.0 | 22.2 | 17.9 | 10.4 | 32.2 | 35.1 | 20.9 |
| NC-Net [45] | 17.9 | 12.2 | 32.1 | 11.7 | 29.0 | 19.9 | 16.1 | 39.2 | 9.9 | 23.9 | 18.8 | 15.7 | 17.4 | 15.9 | 14.8 | 9.6 | 24.2 | 31.1 | 20.1 |
| HPF [37] | 25.2 | 18.9 | 52.1 | 15.7 | 38.0 | 22.8 | 19.1 | 52.9 | 17.9 | 33.0 | 32.8 | 20.6 | 24.4 | 27.9 | 21.1 | 15.9 | 31.5 | 35.6 | 28.2 |
| SCOT [31] | 34.9 | 20.7 | 63.8 | 21.1 | 43.5 | 27.3 | 21.3 | 63.1 | 20.0 | 42.9 | 42.5 | 31.1 | 29.8 | 35.0 | 27.7 | 24.4 | 48.4 | 40.8 | 35.6 |
| DHPF [39] | 38.4 | 23.8 | 68.3 | 18.9 | 42.6 | 27.9 | 20.1 | 61.6 | 22.0 | 46.9 | 46.1 | 33.5 | 27.6 | 40.1 | 27.6 | 28.1 | 49.5 | 46.5 | 37.3 |
| CHM [35] | _49.6_ | _29.3_ | 68.7 | _29.7_ | _45.3_ | _48.4_ | _39.5_ | 64.9 | _20.3_ | _60.5_ | _56.1_ | 46.0 | _33.8_ | _44.3_ | 38.9 | _31.4_ | _72.2_ | _55.5_ | _46.3_ |
| CATs† | 46.5 | 26.9 | _69.1_ | 24.3 | 44.3 | 38.5 | 30.2 | _65.7_ | 15.9 | 53.7 | 52.2 | _46.7_ | 32.7 | 35.2 | 32.2 | 31.2 | 68.0 | 49.1 | 42.4 |
| CATs | **52.0** | **34.7** | **72.2** | **34.3** | **49.9** | **57.5** | **43.6** | **66.5** | **24.4** | **63.2** | **56.5** | **52.0** | **42.6** | **41.7** | **43.0** | **33.6** | **72.6** | **58.0** | **49.9** |

rate of 3e−5 for the CATs layers and 3e−6 for the backbone features are used, which we gradually decrease during training.

## 4.2 Experimental Settings

In this section, we conduct comprehensive experiments for semantic correspondence, by evaluating our approach through comparisons to state-of-the-art methods including CNNGeo [42], A2Net [49], WeakAlign [43], NC-Net [45], RTNs [23], SFNet [26], HPF [37], DCC-Net [17], ANC-Net [27], DHPF [39], SCOT [31], GSF [21], and CHMNet [35]. In Section 4.3, we first evaluate matching results on several benchmarks with quantitative measures, and then provide an analysis of each component in our framework in Section 4.4. For more implementation details, please refer to our implementation available at `https://github.com/SunghwanHong/CATs`.

**Datasets.** SPair-71k [38] provides total 70,958 image pairs with extreme and diverse viewpoint, scale variations, and rich annotations for each image pair, e.g., keypoints, scale difference, truncation and occlusion difference, and clear data split. Previously, for semantic matching, most of the datasets are limited to a small quantity with similar viewpoints and scales [11, 12]. As our network relies on Transformer which requires a large number of data for training, SPair-71k [38] makes the use of Transformer in our model feasible. we also consider PF-PASCAL [12] containing 1,351 image pairs from 20 categories and PF-WILLOW [11] containing 900 image pairs from 4 categories, each dataset providing corresponding ground-truth annotations.

**Evaluation Metric.** For evaluation on SPair-71k [38], PF-WILLOW [11], and PF-PASCAL [12], we employ a percentage of correct keypoints (PCK), computed as the ratio of estimated keypoints within the threshold from ground-truths to the total number of keypoints. Given predicted keypoint $k_{\text{pred}}$ and ground-truth keypoint $k_{\text{GT}}$, we count the number of predicted keypoints that satisfy following condition: $d(k_{\text{pred}}, k_{\text{GT}}) \leq \alpha \cdot \max(H, W)$, where $d(\cdot)$ denotes Euclidean distance; $\alpha$

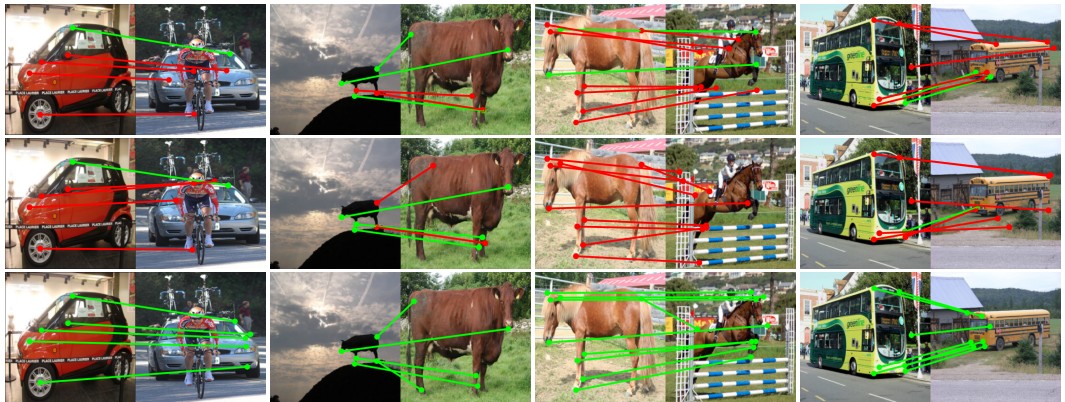

Figure 5: **Qualitative results on SPair-71k [38]:** (from top to bottom) keypoints transfer results by SCOT [31], DHPF [39], and CATs. Note that green and red line denotes correct and wrong prediction, respectively, with respect to the ground-truth.

denotes a threshold which we evaluate on PF-PASCAL with $\alpha_{\mathrm{img}}$, SPair-71k and PF-WILLOW with $\alpha_{\mathrm{bbox}}$; $H$ and $W$ denote height and width of the object bounding box or entire image, respectively.

## 4.3 Matching Results

For a fair comparison, we follow the evaluation protocol of [37] for SPair-71k, which our network is trained on the training split and evaluated on the test split. Similarly, for PF-PASCAL and PF-WILLOW, following the common evaluation protocol of [13, 23, 17, 37, 39], we train our network on the training split of PF-PASCAL [12] and then evaluate on the test split of PF-PASCAL [12] and PF-WILLOW [11]. All the results of other methods are reported under identical setting.

Table 1 summarizes quantitative results on SPair-71k [38], PF-PASCAL [12] and PF-WILLOW [11]. We note whether each method leverages multi-level features and fine-tunes the backbone features in order to ensure a fair comparison. We additionally denote the types of cost aggregation. Generally, our CATs outperform other methods over all the benchmarks. This is also confirmed by the results on SPair-71k, as shown in Table 2, where the proposed method outperforms other methods by large margin. Note that CATs† reports lower PCK than that of CHM, and this is because CHM fine-tunes its backbone networks while CATs† does not. Fig. 5 visualizes qualitative results for extremely challenging image pairs. We observe that compared to current state-of-the-art methods [31, 39], our method is capable of suppressing noisy scores and find accurate correspondences in cases with large scale and geometric variations.

It is notable that CATs generally report lower PCK on PF-WILLOW [11] compared to other state-of-the-art methods. This is because the Transformer is well known for lacking some of inductive bias. When we evaluate on PF-WILLOW, we infer with the model trained on the training split of PF-PASCAL, which only contains 1,351 image pairs, and as only relatively small quantity of image pairs is available within the PF-PASCAL training split, the Transformer shows low generalization power. This demonstrates that the Transformer-based architecture indeed requires a means to compensate for the lack of inductive bias, e.g., data augmentation.

## 4.4 Ablation Study

In this section we show an ablation analysis to validate critical components we made to design our architecture, and provide an analysis on use of different backbone features, and data augmentation. We train all the variants on the training split of SPair-71k [38] when evaluating on SPair-71k, and train on PF-PASCAL [12] for evaluating on PF-PASCAL. We measure the PCK, and each ablation experiment is conducted under same experimental setting for a fair comparison.

**Network Architecture.** Table 3 shows the analysis on key components in our architecture. There are four key components we analyze for the ablation study, including appearance modelling, multi-level aggregation, swapping self-attention, and residual connection.

We first define the model without any of these as baseline, which simply feeds the correlation map into the self-attention layer. We evaluate on SPair-71k benchmark by progressively adding the each key component. From **I** to **V**, we observe consistent increase in performance when each component is added. **II** shows a large improvement in performance, which demonstrates that the appearance modelling enabled the model to refine the ambiguous or noisy matching scores. Although relatively small increase in PCK for **III**, it proves that the proposed model successfully aggregates the multi-level correlation maps. Furthermore, **IV** and **V** show apparent increase, proving the significance of both components.

Table 3: **Ablation study of CATs.**

| | Components | SPair-71k $\alpha_{\mathrm{bbox}} = 0.1$ |
|---|---|---|
| **(I)** | Baseline | 26.8 |
| **(II)** | + Appearance Modelling | 33.5 |
| **(III)** | + Multi-level Aggregation | 35.9 |
| **(IV)** | + Swapping Self-Attention | 38.8 |
| **(V)** | + Residual Connection | 42.4 |

**Feature Backbone.** As shown in Table 4, we explore the impact of different feature back-bones on the performance on SPair-71k [38] and PF-PASCAL [12]. We report the results of models with backbone networks frozen. The top two rows are models with DeiT-B [55], next two rows use DINO [4], and the rest use ResNet-101 [14] as backbone. Specifically, subscript `single` for DeiT-B and DINO, we use the feature map extracted at the last layer for the single-level, while for subscript `all`, every feature map

Table 4: **Ablation study of feature backbone.**

| Feature Backbone | SPair-71k $\alpha_{\mathrm{bbox}} = 0.1$ | PF-PASCAL $\alpha_{\mathrm{img}} = 0.1$ |
|---|---|---|
| DeiT-B$_{\texttt{single}}$ [55] | 32.1 | 76.5 |
| DeiT-B$_{\texttt{all}}$ [55] | 38.2 | 87.5 |
| DINO w/ ViT-B/16$_{\texttt{single}}$ [4] | 39.5 | 88.9 |
| DINO w/ ViT-B/16$_{\texttt{all}}$ [4] | 42.0 | 88.9 |
| ResNet-101$_{\texttt{single}}$ [14] | 37.4 | 87.3 |
| ResNet-101$_{\texttt{multi}}$ [14] | 42.4 | 89.1 |

from 12 layers is used for cost construction. For ResNet-101 subscript `single`, we use a single-level feature cropped at $\mathrm{conv}4 - 23$, while for `multi`, we use the best layer subset provided by [37]. Summarizing the results, we observed that leveraging multi-level features showed apparent improvements in performance, proving effectiveness of multi-level aggregation introduced by our method. It is worth noting that DINO, which is more excel at dense tasks than DeiT-B, outperforms DeiT-B when applied to semantic matching. This indicates that fine-tuning the feature could enhance the performance. To best of our knowledge, we are the first to employ Transformer-based features for semantic matching. It would be an interesting setup to train an end-to-end Transformer-based networks, and we hope this work draws attention from community and made useful for future works.

**Data Augmentation.** In Table 5, we compared the PCK performance between our variants and DHPF [39]. We note if the model is trained with augmentation. For a fair comparison, we evaluate both DHPF [39] and CATs trained on SPair-71k [38] using strong supervision, which assumes that the ground-truth keypoints are given. The results show that compared to DHPF, a CNN-based method, data augmentation has a larger influence on CATs in terms of performance. This demonstrates that not only we eased the data-hunger problem inherent in Transform-

Table 5: **Effects of augmentation.**

| | Augment. | SPair-71k $\alpha_{\mathrm{bbox}} = 0.1$ |
|---|---|---|
| DHPF [39] | ✗ | 37.3 |
| DHPF [39] | ✓ | 39.4 |
| CATs | ✗ | 43.5 |
| CATs | ✓ | 49.9 |

ers, but also found that applying augmentations for matching has positive effects. Augmentation technique would bring a highly likely improvements in performance, and we hope that the future works benefit from this.

**Serial swapping.** It is apparent that Equation 2 is not designed for an order-invariant output. Different from NC-Net [45], we let the correlation map undergo the self-attention module in a serial manner. We conducted a simple experiment to compare the difference between each approach. From experiments, we obtained the results of parallel and serial processing on SPair-71k with $\alpha_{\mathrm{bbox}} = 0.1$, which are PCK of 40.8 and 42.4, respectively. In light of this, although CATs may not support order invariance, adopting serial processing can obtain higher PCK as it has a better capability to reduce inconsistent matching scores by additionally processing the already processed cost map, which we finalize the architecture to include serial processing.

## 4.5 Analysis

**Visualizing Self-Attention.** We visualize the multi-level attention maps obtained from the Transformer aggregator. As shown in Fig. 6, the learned self-attention map at each level exhibits different

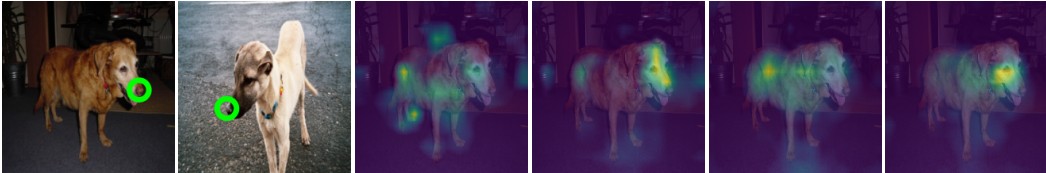

Figure 6: **Visualization of self-attention:** (from left to right) source and target images, and multi-level self-attentions. Note that each attention map attends different aspects, and CATs aggregates the cost leveraging hierarchical semantic representations.

aspect. With these self-attentions, our networks can leverage multi-level correlations to capture hierarchical semantic feature representations effectively.

**Memory and run-time.** In Table 6, we show the memory and run-time comparison to NC-Net [45], SCOT [31], DHPF [39] and CHM [35] with CATs. For a fair comparison, the results are obtained using a single NVIDIA GeForce RTX 2080 Ti GPU and Intel Core i7-10700 CPU. We measure the inference time for both the process without counting feature extraction, and the whole process. Thanks to Transformers' fast com-

Table 6: **Memory and run-time comparison.** Inference time for aggregator is denoted by $(\cdot)$.

|  | Aggregation | Memory [GB] | Run-time [ms] |
|---|---|---|---|
| NC-Net [45] | 4D Conv. | **1.2** | 193.3 (166.1) |
| SCOT [31] | OT-RHM | 4.6 | 146.5 (81.6) |
| DHPF [39] | RHM | 1.6 | 57.7 (29.5) |
| CHM [35] | 6D Conv | 1.6 | 47.2 (38.3) |
| CATs | Transformer | 1.9 | **34.5 (7.4)** |

putation nature, compared to other methods, our method is beyond compare. We also find that compared to other cost aggregation methods including 4D, 6D convolutons, OT-RHM and RHM, ours show comparable efficiency in terms of computational cost. Note that NC-Net utilizes a single feature map while other methods utilize multi-level feature maps. We used the standard self-attention module for implementation, but more advanced and efficient transformer [32] architectures could reduce the overall memory consumption.

### 4.6 Limitations

One obvious limitation that CATs possess is that when applying the method to non-corresponding images, the proposed method would still deliver correspondences as it lacks power to ignore pixels that do not have correspondence at all. A straightforward solution would be to consider including a module to account for pixel-wise matching confidence. Another limitation of CATs would be its inability to address a task of finding accurate correspondences given multi-objects or non-corresponding objects. Addressing such challenges would be a promising direction for future work.

## 5   Conclusion

In this paper, we have proposed, for the first time, Transformer-based cost aggregation networks for semantic correspondence which enables aggregating the matching scores computed between input features, dubbed CATs. We have made several architectural designs in the network architecture, including appearance affinity modelling, multi-level aggregation, swapping self-attention, and residual correlation. We have shown that our method surpasses the current state-of-the-art in several benchmarks. Moreover, we have conducted extensive ablation studies to validate our choices and explore its capacity. A natural next step, which we leave for future work, is to examine how CATs could extend its domain to tasks including 3-D reconstruction, semantic segmentation and stitching, and to explore self-supervised learning.

## Acknowledgements

This research was supported by the MSIT, Korea, under the ICT Creative Consilence program (IITP-2021-2020-0-01819) and (No. 2020-0-00368, A Neural-Symbolic Model for Knowledge Acquisition and Inference Techniques) supervised by the IITP and National Research Foundation of Korea (NRF-2021R1C1C1006897).

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
