# Semantic Correspondence with Transformers
## - Supplementary Materials -

In this document, we provide more implementation details of CATs and more results on SPair-71k [16], PF-PASCAL [4], and PF-WILLOW [3].

## Appendix A. More Implementation Details

**Network Architecture Details.** Given resized input images $I_s, I_t \in \mathbb{R}^{256 \times 256 \times 3}$, we conducted experiments using different feature backbone networks, including DeiT-B [22], DINO [2] and ResNet-101 [5]. For the ResNet-101$_{\texttt{multi}}$ in the paper, we use the best layer subset [15] of (0,8,20,21,26,28,29,30) for SPair-71k, and (2,17,21,22,25,26,28) for PF-PASCAL and PF-WILLOW. We resized the spatial resolution of extracted feature maps to $16 \times 16$. The extracted features undergo $l$-2 normalization and the correlation maps are constructed using dot products. Contrary to original Transformer [23] with encoder-decoder architecture, CATs is an encoder-only architecture. Within our Transformer aggregator, as explained in the paper, we concatenate the embedded features with correlation maps. We feed the resized features into the projection networks to reduce the dimension from $c$ to 128, where $c$ is the channel dimension of the feature. We then feed the augmented correlation map into the transformer encoder, which we use 1 encoder layer and 6 heads in multi-head attention layers. We then use soft-argmax function [8] with temperature $\tau = 0.02$ to infer a dense correspondence field.

**Training Details.** For training on both SPair-71k [16] and PF-PASCAL [4], we set the initial learning rate for CATs as 3e-5 and backbone networks as 3e-6. We then decrease the learning rate using multi-step learning rate decay [18]. We use a batch size of 32. We trained our networks using AdamW [13] with weight decay of 0.05. For data augmentation implementation, we implemented random cropping of image with probability set to 0.5, and used functions implemented by [1] as shown in Table 1.

Table 1: **Data Augmentation.**

|       | Augmentation type       | Probability |
|-------|-------------------------|-------------|
| **(I)**   | ToGray                  | 0.2         |
| **(II)**  | Posterize               | 0.2         |
| **(III)** | Equalize                | 0.2         |
| **(IV)**  | Sharpen                 | 0.2         |
| **(V)**   | RandomBrightnessContrast | 0.2        |
| **(VI)**  | Solarize                | 0.2         |
| **(VII)** | ColorJitter             | 0.2         |

## Appendix B. Reasoning of Architectural Choices

**Correlation Map.** Given the results from ablation study on architecture designs in the paper, we find that use of appearance and the self-attention mechanism are critical to the performance. However, since transformers have the ability to perform dot products and use of appearance is critical for matching task, one may raise a question: *Why correlation map?* As a concurrent work, COTR [6] attempts to omit correlation map and lets transformers to make correlation among features. They show that this is a highly effective strategy in forming correspondences.

Table 2: **Ablation study of correlation map.**

| Method    | SPair-71k $\alpha_{\text{bbox}} = 0.1$ |
|-----------|----------------------------------------|
| CATs†     | 42.4                                   |
| w/o corr. | 37.3                                   |

As shown in the Table 2, we conducted an ablation study to find out if the use of cost volume is beneficial for our setting. We conduct the experiment with the simplest setup by setting the values of correlation map to zeros. In Table 3, for the experimental setting for COTR, we excluded zoom in

35th Conference on Neural Information Processing Systems (NeurIPS 2021)

technique, set the number of layers in transformer to 1 and changed the architecture to output a flow map instead of pixel coordinates. We used single pair of feature maps for computing correlation map and left all other components in the pipeline the same. More details of setting of both experiments can be found in supplementary materials.

Given Table 3 in main paper and the results for experiments validating the use of correlation map, we could say that the sole use of transformer (with its ability to perform dot products) or sole use of appearance is not sufficient, but rather use of both cost volume and appearance allow the transformer to relate the pairwise relationships and appearance, which helps to find more accurate correspondences. However, this is an ongoing research topic whether explicitly using the correlation map for forming correspondences is better or not, which we leave to community for further study.

Table 3: **Comparison to COTR.** *MA.: Multi-level Aggregation*

| Model | SPair-71k $\alpha_{bbox} = 0.1$ | Run-time [ms] |
|---|---|---|
| COTR [6] | 22.1 | 56.1 |
| CATs† w/o MA. | 37.4 | 39.1 |

## Appendix C. Additional Results

**More Qualitative Results.** We provide more comparison of CATs and other state-of-the-art methods on SPair-71k [16], PF-PASCAL [4], and PF-WILLOW [4]. We also present multi-head and multi-level attention visualization on SPair-71k in Fig 4, and multi-level aggregation in Fig 5.

## Broader Impact

Our cost aggregation networks can be beneficial in a wide range of applications including semantic segmentation [19, 21, 14], object detection [10], and image editing [20, 11, 9, 7], as well as dense correspondence. For example, some methods for semantic segmentation tasks require cost volume aggregation. Such adoption would enhance the performance, which could affect various applications, e.g., autonomous driving. On the other hand, our module risks being used for malicious works, which includes image surveillance system, but on its own, we doubt that it can be used for such works.

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

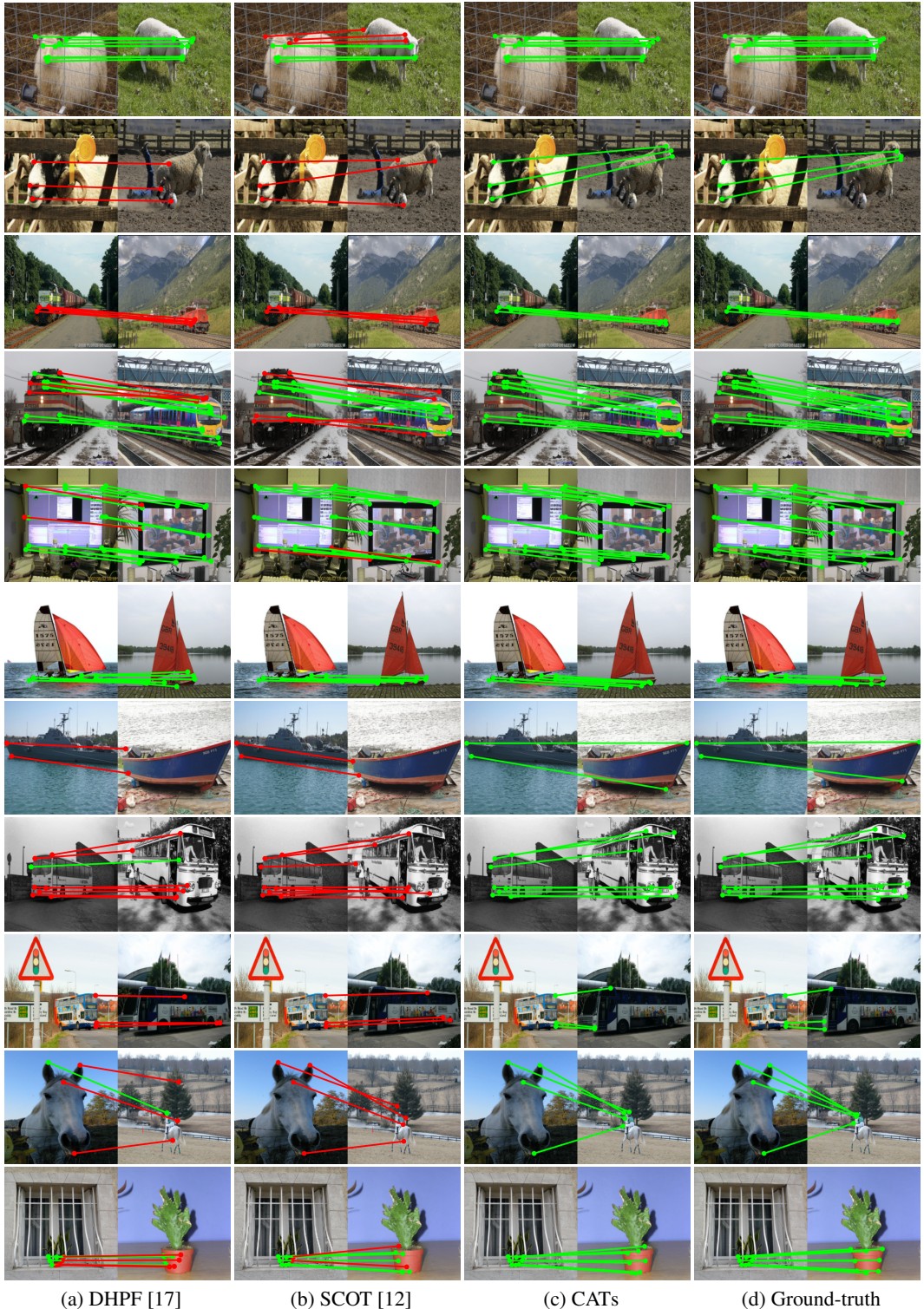

|     (a) DHPF [17]     |     (b) SCOT [12]     |     (c) CATs     |     (d) Ground-truth     |

Figure 1: **Qualitative results on SPair-71k [16]:** keypoints transfer results by (a) DHPF [17], (b) SCOT [12], and (c) CATs, and (d) ground-truth. Note that green and red line denotes correct and wrong prediction, respectively, with respect to the ground-truth.

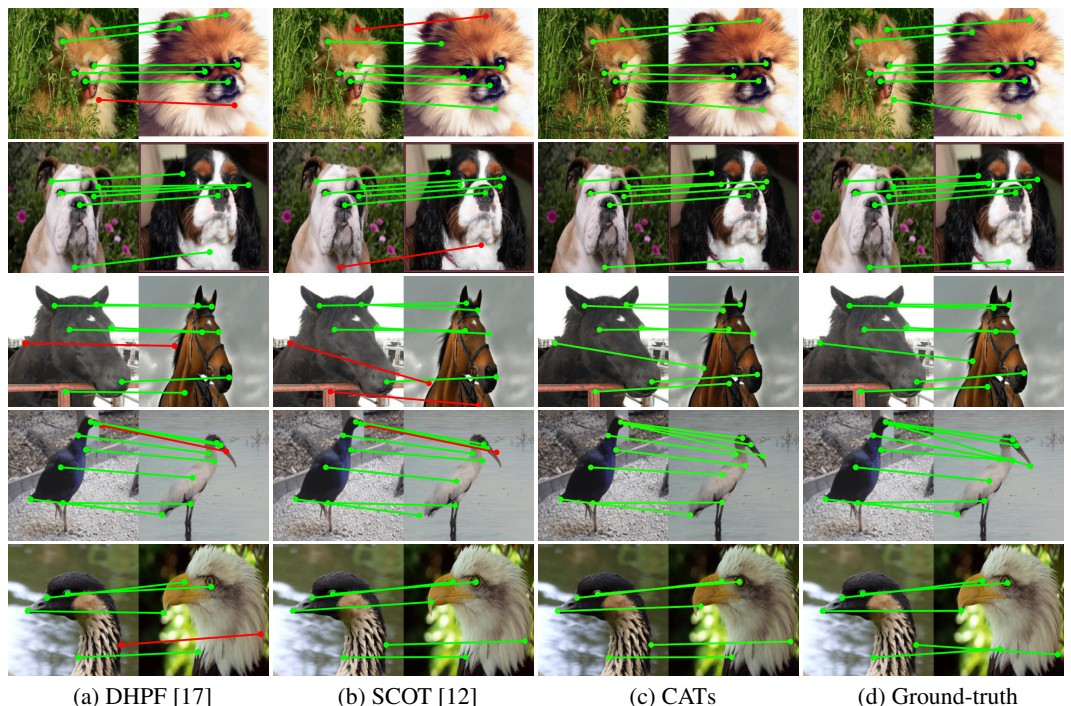

| (a) DHPF [17] | (b) SCOT [12] | (c) CATs | (d) Ground-truth |

Figure 2: **Qualitative results on PF-PASCAL [4]**

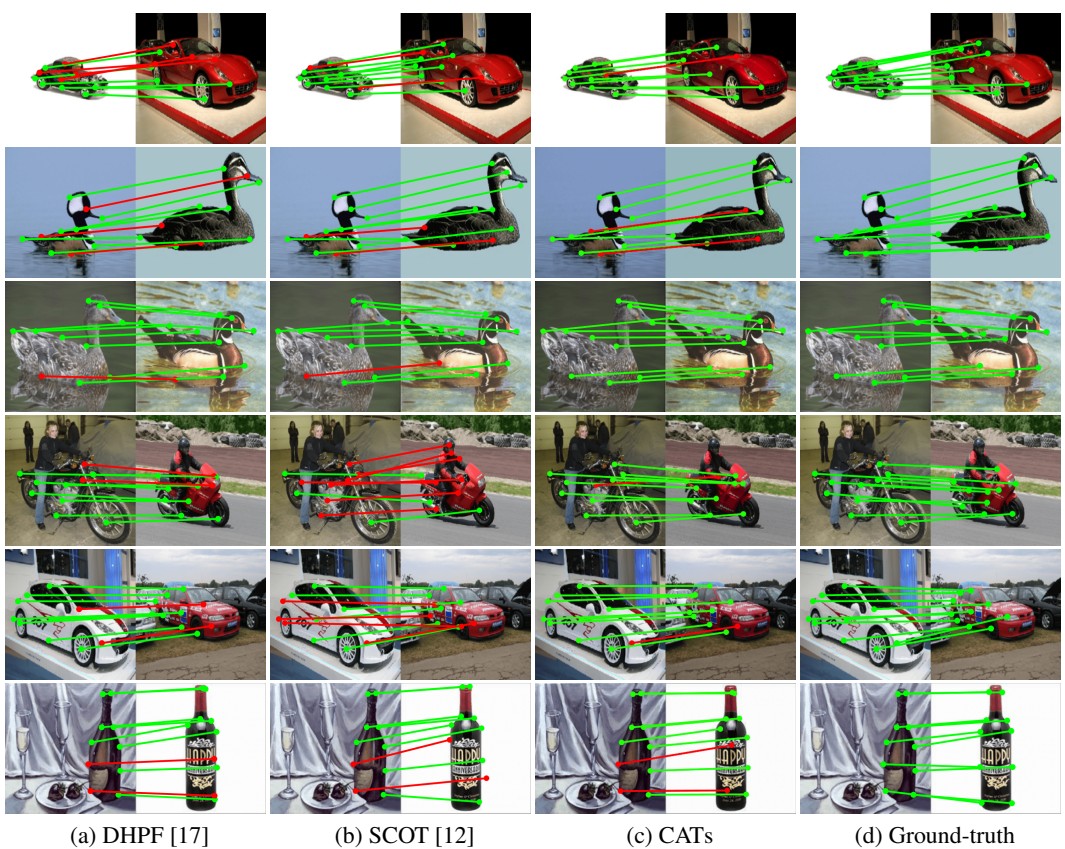

| (a) DHPF [17] | (b) SCOT [12] | (c) CATs | (d) Ground-truth |

Figure 3: **Qualitative results on PF-WILLOW [3].**

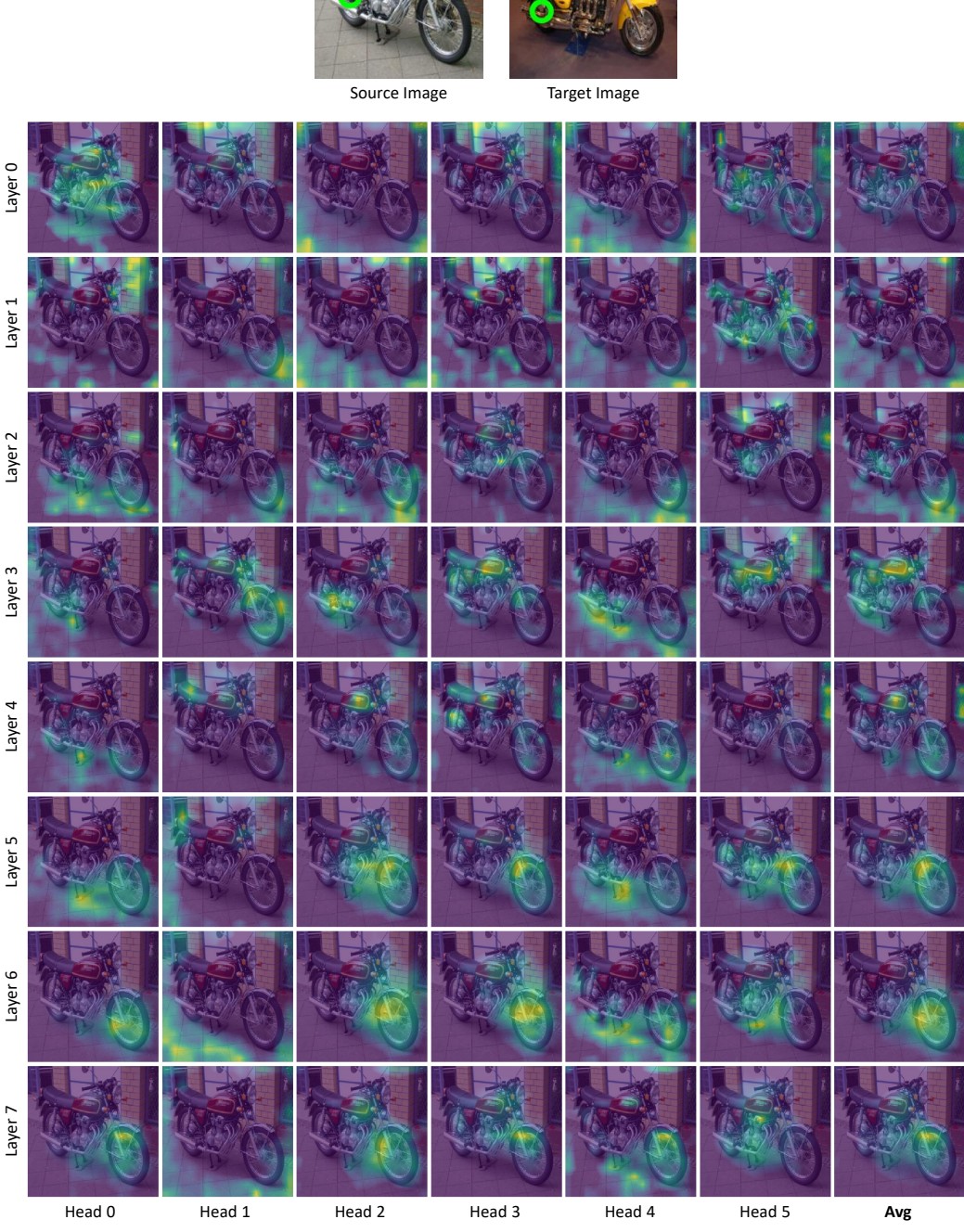

Figure 4: **Visualization of multi-head and multi-level self-attention.** Each head at l-th level layer, specifically among (0,8,20,21,26,28,29,30) layers of ResNet-101 [5] as in [15], attends different regions, which CATs successfully aggregates the multi-level correlation maps to infer reliable correspondences.

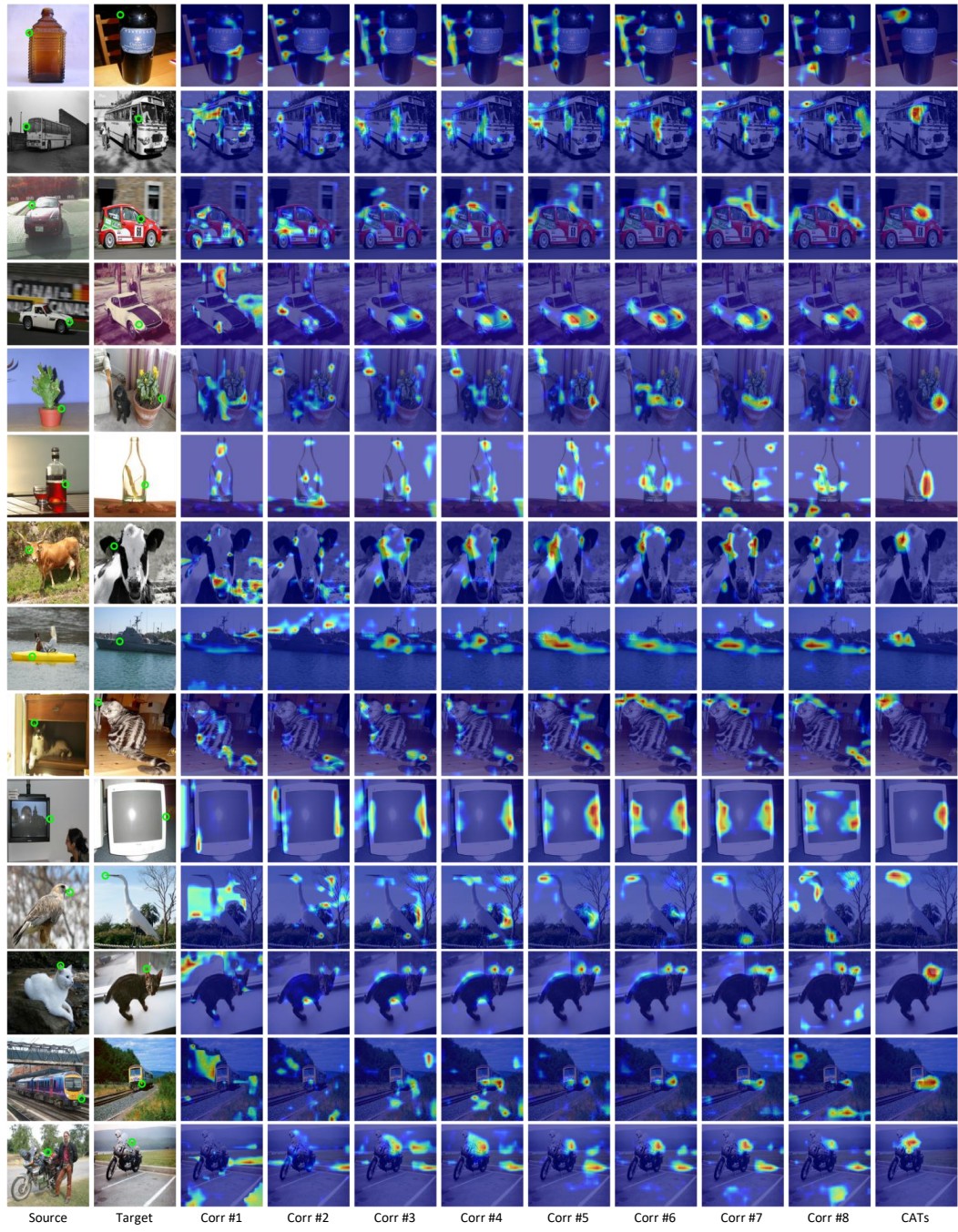

Figure 5: **Visualization of multi-level aggregation.** Each correlation refers to one of the (0,8,20,21,26,28,29,30) layers of ResNet-101, and our proposed method successfully aggregates the multi-level correlation maps.