# OpenReview forum: "CATs: Cost Aggregation Transformers for Visual Correspondence"
_NeurIPS.cc/2021/Conference — NeurIPS 2021 Poster_

### Official Review · Reviewer_TuyT · 2021-07-16

**Rating:** 6
**Confidence:** 3

**Summary:**

The paper deals with the task of semantic correspondence, i.e., finding correspondence between semantically similar images, e.g., the front left paw of two different dogs.  In a nutshell, the paper explores the use of Transformers for this problem and demonstrate that the built-in global receptive field is very beneficial for this task.  Specifically, the authors improve the cost-aggregation part of a semantic correspondence pipeline, which refines the initial matching costs.  The authors propose to concatenate the correlation between pixels (on a feature map from a neural network) with the corresponding appearance embedding, which helps to disambiguate noisy or ambiguous matches (like the left and right eye of a cat as in Figure 2).  Results are demonstrated on standard benchmarks.

**Limitations And Societal Impact:**

I did not find any mentions of limitations, but the broader impact to society is discussed in the supplemental material.

**Main Review:**

### Originality

Although the paper is not ground-breaking, it demonstrates how Transformer modules can be beneficial for the task of semantic correspondence.


### Clarity

The paper is well written, apart from a few typos.


### Experiments

- In Table 1, I think it makes sense to see an improvement for the dataset SPair-71k because it contains more data. But compared to CHM, the performance of the proposed method is not great; a rank-based metric to compare CHM and CATs would actually go in favor of CHM, when only considering columns where both methods have valid results.  This brings me to the next question; why are no results of CHM for PCK threshold of 0.15?


### Minor comments

- Line 4: Please split the sentence "Compared to previous ..." to make it easier to comprehend.
- The abstract is vague and hard to understand without already knowing the typical pipeline for semantic correspondence frameworks.


### Post-rebuttal update

After having read all other reviews and the authors' response, I have decided to keep my initial rating and recommend accepting the submission. However, I also agree with the comments from other reviews on the missing justification of using Transformers and cost volumes. Irrespective of the acceptance decision, I strongly encourage the authors to integrate these additional discussions to make the paper stronger.

**Time Spent Reviewing:**

3.5

---

> ### Author Response · Authors · 2021-08-09
> **Official comment to reviewer TuyT**
>
>     1. Typos
> We thank the reviewer for the comment. We found several typos, e.g., L93, L160 , and we will fix all the typos.
>
>
>     2. Performance of CHM in comparison to CATs
> By the time when we submitted the paper for NeurIPs, only the paper without code implementation became public for CHM. However, currently, the official code is available, and we thus utilized the pretrained weights and codes provided by the authors to reproduce the results. We obtained 94.7% for PF-PASCAL with alpha=0.15 and 90.1% for PF-WILLOW with alpha=0.15.  Comparing the results with those of CATs, we observed that CATs attain 1.7% higher PCK for PF-PASCAL and 0.2% for PF-WILLOW. The results are summarized below:
>
> |Methods|aero.| bike| bird| boat| bott.| bus| car| cat |chai.| cow| dog| hors. |mbik. |pers. |plan.| shee. |trai.| tv |all|
> |:---|:---:|:---:|:---:|:---:|:---:|:---:|:---:|:---:|:---:|:---:|:---:|:---:|:---:|:---:|:---:|:---:|:---:|:---:|:---:|
> |CHM|49.1|33.6|64.5|32.7|44.6|47.5|43.5|57.8|21.0|61.3|54.6|43.8|35.1|43.7|38.1|33.5|70.6|55.9|46.3|
> |CATs &dagger;|46.5| 26.9| 69.1| 24.3| 44.3| 38.5| 30.2 |65.7 |15.9 |53.7| 52.2| 46.7 |32.7| 35.2 |32.2 |31.2| 68.0 |49.1| 42.4 |
> |CATs|52.0| 34.7| 72.2| 34.3| 49.9| 57.5| 43.6| 66.5| 24.4 |63.2| 56.5 |52.0 |42.6 |41.7 |43.0 |33.6 |72.6| 58.0 |49.9|
>
>
> |Model|PF-PASCAL PCK@0&#46;15|PF-WILLOW PCK@0&#46;15|
> |---|:---:|:---:|
> |CHM|94.9|87.5|
> |CATs|96.4|90.3|
>
> Moreover, note that CATs&dagger; in Table 1 reports results with feature backbone frozen, and fair comparison would be comparing CHM with CATs. Given these, we could say that overall performance of CATs is better than that of CHM. We will include this in Table 1.
>
>
>     3. Splitting the sentence in Line 4
> We thank the reviewer for the suggestion. We will fix it. We believe that significant improvements could be made to our paper by proofreading the paper to ensure high-quality presentation.
>
>
>     4. Vague abstract.
> We again thank the reviewer for the suggestion, and we will make an improvement to ensure better presentation as almost all the reviewers pointed out. Specifically, we will include an additional section in the supplementary material to suggest the readers unfamiliar with some of the concepts in our paper to refer. Also any jargon we introduced will be introduced with more sufficient high-level explanation, and make any somewhat exaggerated arguments weaker.

---

### Official Review · Reviewer_FB1A · 2021-07-17

**Rating:** 5
**Confidence:** 5

**Summary:**

The paper proposes a cost-volume refinement method using transformers to be used for a cost-volume-based semantic correspondence framework. The main innovations of the method are the use of transformers, how the cost volume is aggregated with appearance features, and the serial way in which appearance features are used. The method is tested on SPair-71K, PF-Pascal, and PF-Willow datasets.

**Limitations And Societal Impact:**

The paper does not detail the limitations of the proposed method. One obvious limitation that the reviewer can think of would be when applying the method to non-corresponding images. It is likely that the proposed method would still deliver correspondences then. It is also somewhat unclear how the method would perform for pixels that do not have correspondence at all. Recent works, e.g. PDC-Net [57] allows such regions to be ignored. I am not sure the current method is able to do so.

The appendix does discuss broader impact. I would also imagine current work could be used to make deep fakes even more realistic, as a societal impact. Still, the work itself does not have immediate ethical concerns.

**Main Review:**

The paper provides decent performance, outperforming the competitors on SPair-71K, while performing slightly worse in PF-Pascal and PF-Willow. However, as the main argument of the paper is about the use of Transformers for cost volume refinement, this has to be well justified, which the current paper lacks. Considering this, with the presentation quality issues, the reviewer is currently leaning towards rejection. The reviewer, however, would be very happy to be convinced otherwise.

1. Why Transformers?

Contrary to the argument made in the paper, e.g. lines 87--88, CNN-based architectures do not necessarily suffer from local receptive fields. For example, even the featuremap in this paper, which is 16x16 at the end, would have a receptive that roughly covers the entire image. For example, resnet 101 (v1) has a receptive field size of 1027 pixels. This, combined with the post-processing or hierarchical processing makes the locality argument less convincing. Moreover, a naive way to remove the locality constraint for a 16x16 feature map would be to use MLPs.

In fact, a critical component of the transformer is that they actually relate and route features according to the pair-wise relationships -- the core idea behind the self-attention module. Hence I would suggest that the authors reconsider this argument.

Given this, what is most important in this paper is in fact, not just achieving state-of-the-art performance, but rather revealing that one should for sure head towards transformers. The architecture suggested in this paper is a perfect playground for this as the T function could be replaced with any deep network -- which is quite rare! One could make use of CNNs, (both 2D and 3D), as well as a fully connected layer. This ablation study is a must for the paper to deliver its main message, but is missing. Without a clear justification on why transformer is necessary, the contributions of the paper itself are quite incremental, as many of the components are similar to existing work (e.g. [35]). The reviewer would like to note that the ablation study in the current manuscript is great, but they are somewhat tangent to showing that the use of a transformer is essential.

2. Why cost volume?

As shown in Table 3, it seems that the use of appearance, and how the transformer utilizes it seems to be critical to the final performance. However, since transformers have the ability to perform dot products, at this point, it becomes somewhat questionable whether the cost volume itself is necessary at all. For example, what would happen if everything in the pipeline remains the same but the cost volume is removed? This may create some issues related to connecting between the two images, but then the serial way in which the appearances of the two images are provided may allow information to flow through and let the transformer perform this matching by itself. Hence this is perhaps another ablation study that is required to justify the architecture.

3. Missing reference

Following up, in fact in COTR [a], the paper does something along these lines where cost volume is omitted and it is left to the transformer to make correlations among features. Their work shows that this is a highly effective strategy in forming correspondences. This work is also the first work that the reviewer is aware of on applying Transformers to the correspondence problem (their work is not on semantic correspondences though), predating also LoFTR in terms of the date it became public. While it is probably unreasonable for the paper to compare against this method as it is aiming for geometric correspondences (COTR code page does show a demo for semantic correspondence across human faces), the paper should at least differentiate itself from this method.

[a] Jiang et al., "COTR: Correspondence Transformer for Matching Across Images", ArXiv 21

4. Serial  swapping

The serial swapping suggested in Eq 2 is not order invariant. Hence, differently from NC-Net, depending on which image is used first, it will give different outcomes. Is there some mitigation strategy to avoid this from happening? Otherwise, this does not seem like a proper way to encourage cycle consistency, but another way to incorporate the two appearances.

5. Presentation

The quality of the presentation requires improvement. The paper does not read well and has multiple grammatical errors that harm the paper's quality. For example, "appearance affinity modeling" in line 9 is unclear, as well as lines 11--14 unless the reader has already read the paper. In line 26, the paper states that it has been proven that more powerful feature representation... but this is just supported by empirical evidence. A similar mistake is in line 202, where the paper states augmentation guarantees performance boost, which is not always true. Other examples include; In line 22, "unconstrained settings" is unclear; line 40 "formulated variously"; line 65 "without a means to refine..." phrase seems discontinued from the previous phrase; line 131 "by first feeding into scaled dot product attention function" object of the sentence is missing; line 199 "inductive bias" could be anything and not clear as Transformers also have inductive bias coming from their particular structure; line 204 "have empirically found" should be present tense to be consistent; line 218 no need for "basically".

There are some organizational issues as well, for example, lines 104--115 are somewhat repeating what is said in the related works section, and it is the reviewer's personal opinion that they belong also either in the intro or related works as they are not directly relevant to the method. Similarly, 130--136 also discusses Transformers and Layer Normalization in general, which is again more suited in the related works section.

Many important details about the method are also left to reference. For example, in lines 119--121, the paper refers to 35 for how multi-level features are extracted, but without this information, it is hard to understand the paper. For completeness, this information should be included. It seems that some of the details are present in the supplementary appendix, but in this case, the main paper should say so. Another example where detail is missing is related to augmentation in lines 206--207, as without reading [6], it's impossible to replicate the paper.

6. Promises in the checklist are not delivered

1(c) should be Yes, with a pointer to the appendix.

3(d), 4(a), 4(b) are promised, but not provided.

==== Post Rebuttal Update ====

I am convinced by the new experiments that the reviewers have added, and agree that it is now enough to empirically justify the proposed method's design. However, I am concerned that incorporating these changes would require a significant amount to rewrite, which we have no means of verifying its quality after edit. I thus still vote for rejecting the paper. It seems like the submission was an unfinished product, that was completed during the rebuttal period.

**Time Spent Reviewing:**

4

---

> ### Author Response · Authors · 2021-08-09
> **Official comment to reviewer FB1A  -   1**
>
>     1. Why transformers?
> We are greatly thankful for this constructive comment. Indeed with CNNs, the receptive can cover any size of input given sufficiently stacked convolutional layers. For example, when addressing a correlation map of size (16x16) x (16x16), utilizing CNNs can definitely cover the whole map with only a few stacked layers. However, although CNN-based methods can handle any arbitary size of input, transformers can *explicitly* consider pairwise relationships, learn long-term dependency and process the input as a whole while conventional CNN-based methods require a number of convolution operations to cover the whole input. This forces them to *implicitly* learn the relationships (activation maps after each convolutional layer contain information extracted from past activation maps), which could significantly differentiate the information extracted by transformers and CNN-based [a].  Moreover, although a naive way to remove the locality constraint would be to use MLPs as the reviewer pointed out, using simple MLP may lack capability to fully consider pairwise relationships compared to Transformers [b].  We will include this when addressing the argument (CNNs have limited receptive field).Taking this into account, we would like to then talk about the necessity of use of transformers.
>
> We agree that the T function within our framework could be replaced by CNNs, MLPs or any other method that is capable of relating the features. We thus compared CATs with all the other cost aggregation methods in Table 1 to validate the use of Transformers and show its superiority to others. However, the paper indeed lacked an ablation study to *explicitly* justify the reason to use transformers over other methods as COTR has done by replacing transformer with MLP. Hence, given this and the argument we made about transformers can *explicitly* learn relationships while CNNs *implicitly* learn, we conducted an additional ablative experiment to *explicitly* validate the reason to choose transformers over other methods as the reviewer suggested. We conducted two experiments: one for replacing the Transformer within our framework with pure MLP that only does channel-mixing operation; the other one for replacing the Transformer with MLP-Mixer, a module that includes both channel-mixing and token-mixing operation to approximate pairwise relationships. We conducted the experiment in which all other components in the pipeline remained the same. The results are shown below:
>
> |Aggregation|SPair-71K PCK@0&#46;1|
> |:---|:---:|
> |MLP|34.4|
> |MLP-Mixer|39.1|
> |Transformer|42.4|
>
> Using simple MLP yields surprisingly competitive results as shown in the Table above and the ablation study COTR conducted, but its relatively poor performance highlights the importance of pairwise relationships. Using MLP-Mixer yields highly competitive results as well, but compared to CATs, the performance is relatively poor. Given this and as both MLP and transformer can remove locality constraint, we then could argue that the transformer better considers the pairwise relationships than MLP and shows its superiority to better take global context in a correlation map into account.
>
> Furthermore, though not the strictly fairest comparison, the results in Table 1 show that transformer outperform CNN-based methods, and combined with the statement about Transformer *explicitly* learns relationships while CNNs *implicitly* learns, we could say that the way each method aggregates the cost volume and the output obtained from each aggregation method significantly differs. With such differences, CATs obtaining better performance over other methods indicates its superiority and advantage to *explicitly* learn pairwise relationships with global attention range, process the input as a whole, and remove locality constraint.  We will include these in the paper along with the implementation details and the result of MLP and MLP-Mixer on PF-PASCAL and PF-WILLOW.
>
> Note that although we couldn’t conduct experiments for CNNs under fair setup (NC module utilizes a single-level feature, which makes multi-level aggregation and appearance affinity modelling infeasible), in Table 4 , CATs obtained 37.4% for SPair-71k with a single feature map, and comparing this with performance by NC-Net (not completely, but almost fair), we could say transformer showed its advantages over CNN-based method. Lastly, we would like to highlight that solely leveraging Transformer does not yield competitive results as our ablation study (Table 3) shows. We believe that aspects, which each component (appearance modelling, multi-level aggregation, swapping self-attention and residual connection) contributes to apparent improvement in performance and our method could be adopted in other domains where cost volumes are utilized, providing a more valuable message than just simply leveraging transformers in this task.
>
>     2. Why cost volume?
> The reviewer asked what would happen if the pipeline remained the same, but the cost volume was removed. In many dense correspondence tasks, it is well known that the sole use of matching cost without a way to aggregate to find correspondences between challenging image pairs is insufficient to find the accurate correspondences. To this end, many works, e.g., NC-Net, SCOT, or DHPF, propose a means to aggregate the cost volume. In this manner, as the cost volume is closely related to our framework and we focus on aggregating the cost volume, it is probably infeasible to exclude the cost volume from our pipeline. In addition, it is well known that use of appearance as an affinity is one of the key components in conventional cost aggregation as shown in many works in stereo matching [c]. That is why we utilize the appearance feature as an additional input to the transformer. Cost aggregation has been adopted in many conventional matching networks, and other domains including semantic segmentation and object detection. Because we proposed a general method in which CATs can be adopted in tasks that need cost aggregation, we would like to stress its general applicability to other domains.
>
> On the other hands, a recently released paper COTR
> proposes to find the correspondences without use of cost volumes.
> It attained state-of-the-art performance in dense correspondence task, demonstrating
> the unnecessity of use of cost volume. However, it remains arguable whether using
> the cost volume is better or not using it is better. We thus conducted an experiment
> that could justify the use of cost volume. This ablation study aims to
> find out if the use of cost volume actually helps to find the correspondences.
> Experimental setting is that surprisingly, making small changes to the
> architecture of COTR allows fair comparison between the architecture that uses
> cost volume and the one that does not. As COTR leverages only the extracted
> features to find the correspondences without use of cost volume, directly comparing COTR with CATs can justify the use of cost volume.
>
> For COTR, we excluded zoom in technique and changed the architecture to output a flow map instead of pixel coordinates. We left all other components in the
> pipeline the same. To make a fair comparison, we set the number of layers
> within encoder and decoder of COTR and encoder module of CATS to be equal, trained on SPair-71k with same number of epochs and learning rate, froze the feature backbone, used a single feature map, and other implementation details will be included in the supplementary material. Note that the default setting for # of
> layers for the encoder and decoder module in COTR was 6 according to the
> released code and the default setting for CATs is 1. The results are shown below:
>
> |Model|# of Layers|SPair-71K PCK@0&#46;1|Inference Time [ms]|
> |---|:---:|:---:|:---:|
> |COTR|1|22.1|56.1|
> |CATs&dagger; w/o multi-level aggregation|1|37.4|39.1|
> |COTR|6|45.0|67.2|
> |CATs&dagger; w/o multi-level aggregation|6|46.9|47.4|
>
> Interestingly, not only we observed significant performance improvements for CATs when the number of layers is set to 6, but also CATs outperformed COTR by a large margin when the number of layers was set to 1. Specifically, when the number of layers is set to 6, performance of CATs improved from 37.4 to 46.9, demonstrating potential of other settings (e.g., feature backbone finetuned).  Given this and Table 3, we could say that the sole use of transformer (with its ability to perform dot products) or sole use of appearance is not sufficient, but rather use of both cost volume and appearance allow the transformer to relate the pairwise relationships and appearance, which helps to find more accurate correspondences. Nevertheless, this topic requires further study as these experiments may not completely find the answer to the use of cost volume. We will include and stress this in the paper as this seems highly important. In addition to the experiments above, we will conduct an ablation study on the influence of the number of layers on performance as well as computational costs, and include this in the supplementary material.
>
>
>
> [a] Naseer et al., "Intriguing Properties of Vision Transformers", arxiv’21
>
> [b] Tolstikhihn et al., “MLP-Mixer: An all-MLP Architecture for Vision”, arxiv’21
>
> [c] Kendall et al., “End-to-End Learning of Geometry and Context for Deep Stereo Regression”, arxiv’17

---

> > ### Comment · Reviewer_FB1A · 2021-08-11
> > **Quick additional questions**
> >
> > Thanks for the thorough reply. Highly appreciated!
> >
> > I have a follow-up question. (Not sure if the authors can still reply though).
> >
> > 1. why transformer
> >
> > Can you elaborate why
> >
> > >Note that although we couldn’t conduct experiments for CNNs under fair setup (NC module utilizes a single-level feature, which makes multi-level aggregation and appearance affinity modelling infeasible),
> >
> > After all, $\mathcal{T}$ could be any function, for example, a 3D CNN for the cost volume. Is there anything stopping this from happening? I think this is probably somewhat minor, given that you already show that it's better than MLP or MLP-mixer.
> >
> > 2. why cost volume
> >
> > Is there a reason why you are not directly removing the cost volume? For example, simply setting it to ones for all values would allow you to easily test this. The experiment that you've done here seems much more complicated (highly appreciated though)
> >
> > Thanks!

---

> > > ### Author Response · Authors · 2021-08-12
> > > **Response to additional questions**
> > >
> > > Thanks for the quick reply!
> > >
> > > We apologize for the late reply. It took us a day to reply as we conducted several experiments to reply with the highest quality.
> > >
> > >     1.	Elaborate why we couldn’t conduct experiments for CNNs under fair setup.
> > > We thought that directly replacing the T function with CNNs may lead to unfairness. To preserve the main components within our architecture, we had to replace only the *self-attention* module within the function T without making any change to the proposed contributions (inter-, intra- and appearance affinity). We can easily do this for MLP or MLP-Mixer, but CNNs require quite an effort. For example, we can’t directly replace self-attention module with 4D convolutions due to the appearance embedding concatenated to the input cost volume [NC-Net]. Similarly for 3D convolutions, the different structure of spatial axis makes them infeasible to directly replace the module. However, now that we realize, we may be able to use 2D CNN for fair comparison by replacing self-attention module without touching our contributions. We will include this in the final. Afterall, although it results abandoning our contributions, we believe that changing the T itself to 3D convolution [a], as the reviewer suggested, allows us to evaluate the performance of “Transformer Aggregator”. We thus conducted an experiment, and the results are shown below:
> > >
> > > |Aggretator|SPair-71K PCK@0&#46;1|
> > > |:---|:---:|
> > > |3D Conv.| 30.6 |
> > > |Transformer|42.4|
> > >
> > > We observed that 3D convolution allows a satisfactory result, but compared to Transformer, the gap seems large. Given this, we evaluated the performance of “Transformer Aggregator” compared to “CNN Aggregator” and transformer showed its superiority to successfully aggregate the cost volumes.
> > >
> > >     2.	Reason for not directly removing the cost volume
> > > The reasons for why we chose to conduct experiments to compare COTR and CATs were that first, when we were discussing about how we could justify the use of cost volume, we thought that directly comparing with COTR provides fair comparison and support justification to the use of cost volume because COTR attains SOTA performance by successfully finding correspondences without the use of cost volumes. The reason for why we did not directly remove the cost volume by setting it to ones for all values was that we simply couldn’t think of this simple idea. However, now that we know it, we conducted experiment to see what would happen with this setup. The results are shown below, which also supports the claim we made in the first rebuttal!
> > >
> > > |Method|SPair-71K PCK@0&#46;1|
> > > |:---|:---:|
> > > |W/o cost volmue | 37.3 |
> > > |W/ cost volume|42.4|
> > >
> > >
> > > We thank the reviewer for constructive comments, we believe that all these discussions deserve a section in supplementary material! We will combine these with the answers in the first rebuttal and include them in the paper!
> > >
> > > If the reviewer has any other question, please let us know!
> > >
> > >
> > > [a] Tran et al., “A closer Look at Spatiotemporal Convolutions for Action Recognition”, CVPR’18

---

> ### Author Response · Authors · 2021-08-09
> **Official comment to reviewer FB1A - 2**
>
>     3. Missing reference
> We will include COTR in the related works as a concurrent work to ours. There are several differences between COTR and CATs. First, COTR does not utilize cost volume for finding correspondences, while conventional matching networks (e.g., NC-Net) do. Regarding the comment above, although the question of what is better still remains unsolved, the experiments showed that the use of a cost volume allowed better performance. Second, it takes each coordinate as an additional input and requires feed-forwarding several times for inference while CATs takes a pair of images and requires a single feed-forward. Third, while CATs outputs a dense flow field, COTR outputs a correspondence between a pair of points and the randomly sampled points during inference could affect the results.
>
>     4. Serial swapping
> We agree that the architectural design of CATs is not order invariant. To explain the reason why we chose to process serially rather than parallely, we compared the results of both on SPair-71k. For parallel processing, we followed the same setting adopted in NC-Net. The results are shown below:
>
>
> |Model Setup|SPair-71K PCK@0&#46;1|
> |---|:---:|
> |Parallel|40.8|
> |Serial|42.4|
>
>
> The reasoning behind choosing serial processing over parallel(NC-Net) was that as shown in the Table above, although CATs is not order invariant, serial processing allowed higher PCK as this has a better capability to reduce inconsistent matching scores. However, we aim to preserve both performance and order invariance, hence we could perhaps combine both serial and parallel processing. Maintaining the serial processing architecture we proposed, we could add another line in a parallel manner to ensure order-invariance.  We believe that it would ensure both order invariance and high performance, and we will validate this in the paper.
>
>     5. Presentation
> We thank the reviewer for the elaborate comments. We will proofread the paper to minimize the typos, grammar errors and any poorly presented statement. Specifically, we will make some of the statements weaker (e.g., augmentation guarantees, powerful feature representation…), provide sufficient explanation for jargons (e.g., appearance affinity modelling), and minimize the grammar errors (e.g., in L204, have empirically found..).
>
>     6. Promises in the checklist are not delivered
> We will include them in the supplementary material. We will read and check again to make sure that every point in the checklist is properly answered.
>
>     7. The paper does not detail the limitations of the proposed method.
> We will create a section in the supplementary material to mention the limitations of the proposed method. We agree that CATs does not consider uncertainty, making it hard to process non-corresponding data. In addition, given multi-objects in an image, it may perform badly, which is also an apparent limitation. However, the results of experiments showed potentials to find accurate correspondences for highly cluttered scenes, and we believe our method, which has shown significant improvement in the challenging correspondence task, can be extended to address multi-object correspondence tasks. We will include this in the newly created section and we are greatly thankful for suggesting interesting directions for future work.

---

### Official Review · Reviewer_CGvU · 2021-07-17

**Rating:** 6
**Confidence:** 4

**Summary:**

In this paper, the authors propose transformer-based cost aggregation networks for semantic correspondence with challenges of large intra-class appearance and geometric variations. The overall learning pipeline hierarchically aggregates the matching scores computed between disambiguated input features from semantically similar images by combining with swapping self-attention and residual connections. Extensive experimental results on three datasets, SPairk-71k, PF-PASCAL, and PF-WILLOW, show its efficacy on finding dense semantic correspondences.

**Ethical Concerns:**

None.

**Limitations And Societal Impact:**

I would like to see how the author would respond to the weaknesses in the previous question and the opinions of other reviewers.

**Main Review:**

[Motivation and reasoning about transformer]
First, the motivation of the proposed system is described clearly, considering the necessity of leveraging transformer-based cost aggregation for dense correspondence estimation between semantically similar images. Overall, the flow of the paper is easy to understand, and the goal is straightforward.

[About architectural design – aggregator]
The overall network architecture consists of feature extraction, cost aggregation, and flow estimation steps. Specifically, on the stage of cost aggregation, each feature is transformed with the proposed module, CAT, and aggregated with intra- and inter-correlation self-attention mechanism. Here, I have a question about the role of each self-attention module. Although in L165, the operation of each module is described well, the motivation for disentangling each module is not clear. I would like to see the reasoning of their design, and what happens if the modules are not decomposed.

[Natural effectiveness of cost aggregation itself]
Inherently, the cost aggregation by inner product has the capability of finding correlation between two input features. As described in Figure 2 (c), the raw correlation map shows its strong capability of finding semantically similar features. Therefore, these results raise a doubt as to whether the proposed CAT module is effectively designed, considering the complexity of the module. I wonder if the effect of CAT can be maximized when finding semantic correspondence for cluttered scenes with high complexity, such as traffic driving environments. Currently, most of the demonstrated dataset images contain only a single object.

[Minor comment]
The title is too ambiguous to deliver/represent the contribution of the works.

### UPDATE AFTER REBUTTAL ###
I appreciate the authors' feedback and valuable comments from other reviewers. Parts of my main concerns (e.g., reasoning about the design of the aggregator) are eased with the authors' fair reasonings. However, as commented by FB1A, those ablation studies (including the justification of transformers and cost volume) need to be discussed in the main paper. In addition, It would be better to include some failure cases or qualitative results with multiple objects for further discussion (although I appreciate the reply from the authors for the issue of multiple objects). Overall, I update my final score from 7 to 6, but still leaning towards acceptance.


**Time Spent Reviewing:**

3

---

> ### Author Response · Authors · 2021-08-09
> **Official comment to reviewer CGvU**
>
>     1. About the architectural design – aggregator: see the reasoning of their design, and what happens if the modules are not decomposed
>
> We are greatly thankful for all the constructive comments that could help make our paper stronger. Inspired from hypercorrelation-squeeze [a] and CoaT [b], as [a] attempted to fully leverage the multi-level correlation and [b] decomposed the attention operations, we speculated that leveraging multi-level correlation and allocating separate (weights are not shared) transformer for *inter*- and *intra*- rather than sharing the weights for both operations would not only help to better capture the rich semantics present in features but also differentiate level-wise and spatial-wise information.
>
> Additionally, the motivation for disentangling each module was that given a combined module without decomposing, perhaps one way to implement it would be tokenizing for all the spatial locations across all the levels $L$. This would cause huge computational costs with complexity of $O(N^2)$ for self-attention, where $N$ indicates number of tokens. As $L$ increases, it gets infeasible to handle the computational cost. Intuitively, because for each $l$-th level, different information is present in that correlation maps are obtained from features from different levels and each spatial location refers to similarity between particular pairs of points, separate modules should address them to effectively capture information from both across in level-wise and spatial-wise manners. As shown in Table 3 (III) , this allowed apparent performance improvements.
>
> To summarize, computational cost and performance were what prevented us from combining both modules, and the motivation was to allocate different module for aggregating across level (*inter*-) and spatial locations (*intra*-). We will include this in the method section where both operations are explained and in the ablation study where Table 3 is analyzed.
>
>     2. The raw correlation map shows strong capability. Whether the proposed CAT module is effectively designed considering the complexity of the module.
>
> Although the raw correlation map showed its capability of finding semantically similar features, we believe that in Figure 2 (c), the raw correlation map is susceptible to repetitive patterns (in this case, eyes).  As (e) shows, our method addresses such a challenge well, successfully locating the right eye. Also, in Figure 4, we visualized the raw correlation maps and outputs after cost aggregation by HPF and CATs. It is apparent that CATs successfully selects the correct wheel of the bicycle given a noisy correlation map as shown in (c) and (d).
>
> In fact, solely leveraging the raw correlation map without post-processing is the Winner-Takes-All method, and we report its results at the top row in Table 1. The result shows relatively poor performance and similar results are reported in SCOT, and this motivates the use of the cost aggregation module. For the complexity of the module, we include a table reporting results of an additional ablation study, the same table we presented for reviewer MLbN. As shown below, we find that the complexity is not a critical issue for CATs.
>
> ||Aggregation|Memory [GB]|Inference Time [ms]|
> |------|---|:---:|:---:|
> |NC-Net|4D Conv.|1.2|193.3(166.1)|
> |SCOT|OT-RHM|4.6|146.5(81.6)|
> |DHPF|RHM|1.6|57.7(29.5)|
> |CHM|6D Conv.|1.6|47.2(38.3)|
> |CATs|Transformer|1.9|34.5(7.4)|
>
> Given computational costs and performance improvements made by the CATs module, we believe that CATs shows superiority to other methods.
>
>     3. Multi-object semantic correspondences
>
> SPair-71k is the most recently released large-scale benchmark for semantic correspondence. This challenging dataset contains image pairs with diverse variations in viewpoints and scales and includes background clutters. As CATs attains a highly competitive performance for SPair-71k, we could say that it demonstrated its potential to find accurate correspondences even for cluttered scenes with high complexity.
>
> It should be noted that we utilized standard benchmarks commonly used for semantic correspondence, which include PF-PASCAL and PF-WILLOW, and they mostly contain a single object per image. Indeed this leads to apparent limitations which existing methods (NC-Net, CHM, or GSF) also inherit. Including CATs, these methods focus on finding correspondences given a single object and we will add a section to mention the limitation in the supplementary material. We speculate that given multi-objects, our method may find more accurate correspondences compared to other methods but it is also possible that CATs may fail to find the correct correspondence.
>
> Very recently, as a concurrent work,  DiscoBox [c] attempted to overcome such limitations by jointly learning instance segmentation and semantic correspondence using bounding box supervision. Such joint learning allows guidance to tackle multi-object correspondence problem, which is one of the possible solutions to address the multi-object correspondence task.  As CATs focuses on aggregating the cost volume, it can replace the optimal transport solver used in DiscoBox, demonstrating its potential to address multi-object correspondence tasks. We will include this analysis in the limitation section. We also agree that addressing such a problem is a must direction that methods dealing with semantic correspondence should consider, we thank the reviewer for suggesting a promising direction for future work.
>
>     4. Title too ambiguous
>
> We thank the reviewer for the suggestion, as we also agree that the title is too ambiguous as our contribution is mostly about cost aggregation. So we would like to change our title to “CATs: Cost Aggregation Transformers for Semantic Correspondence”.
>
> [a] Min et al., “Hypercorrelation Squeeze for Few-Shot Segmentation”, ICCV’21
>
> [b] Xu et al., “Co-Scale Conv-Attentional Image Transformers”, arxiv’21
>
> [c] Shiyi et al., “DiscoBox: Weakly Supervised Instance Segmentation and Semantic Correspondence from Box Supervision”, ArXiv 21

---

### Official Review · Reviewer_MLbN · 2021-07-18

**Rating:** 7
**Confidence:** 3

**Summary:**

This paper addresses the problem of finding dense correspondences between semantically similar images. Given a feature extractor, the primary focus of this work is on the design of a matching algorithm to find the optimal matches for each spatial location. The proposed method is a learned transformer based architecture that takes advantage of multi-scale features to resolve ambiguous local matches. Results demonstrate that the performance of the proposed model is comparable to existing methods on multiple datasets.

**Limitations And Societal Impact:**

The authors have not discussed the societal impact of this work.

**Main Review:**

- The paper addresses an important problem of finding correspondences across semantically similar images. Majority of the research in this direction focuses on improving feature extraction methods. However, this work shows that building smarter post-processing algorithms for finding matches could bring significant improvements in the quality of correspondences.

- The proposed method of using a transformer-based architecture for computing feature correlation scores is technically sound, intuitive and novel. The motivation for the presented design of the architecture has been explained thoroughly. The idea of including multi-scale appearance features and processing the intra-level and inter-level features sequentially is also novel and could be adopted in other domains.

- The experimental results show that the proposed model is at least as effective as existing state-of-the-art models for finding semantic correspondences. The qualitative results presented provide insight into the functioning of the self-attention mechanisms and demonstrate that the model can generate high-quality correspondences. Furthermore, the ablative studies show that all the design choices in the transformer-based architecture lead to improved correspondences.

## Concerns/Suggestions
- The quality of writing of the paper could be significantly improved. Especially the abstract and introduction are very difficult to follow. For example, Line 48-56 uses too much jargon without any context provided. It only makes sense after reading the approach section. So I think the paper would benefit from thorough proof-reading.

- Since the proposed cost-aggregation method relies on a transformer-based architecture, it would be good to know the computational cost compared to existing methods like CVF.



**Time Spent Reviewing:**

2.5 hours

---

> ### Author Response · Authors · 2021-08-09
> **Official comment to reviwer MLbN**
>
>     1. The quality of writing of the paper could be significantly improved.
>
> We would like to first thank the reviewer for the constructive comments. We agree that both abstract and introduction need improvements as they barely provide high-level explanation, making it hard for the readers to understand what each jargon represents unless they read the method section as the reviewer points out. We will improve the quality of presentation throughout the paper. Specifically, we will fix, for example, L9-L13 and L48-L56 talking about each module in CATs without any helpful contextual information. Also, we will provide a brief and easy-to-understand high-level explanation before using each jargon (e.g., appearance affinity modelling, multi-level aggregation). In addition, we will include more details to let the readers unfamiliar with the concepts (e.g., self-attention of Transformers) refer to the supplementary material.
>
>     2. It would be good to know the computational cost compared to existing methods like CVF
>
> We agree that it would be very interesting to compare our aggregation method to existing methods as the reviewer mentioned. However, directly comparing our method to those prior to deep learning (e.g., CVF) is probably infeasible due to several reasons (e.g., image size, use of GPU, incompatible library). As an alternative, we included direct comparison to other methods that utilize different aggregation methods (In Table  6, NC-Net, SCOT, DHPF), and we observed that our method has an apparent advantage in terms of run-time. As our method is based on transformers, it inherits transformers’ advantage of fast inference speed thanks to parallelizable nature[a]. We will include run-time of CHM in the final version, as the code and weights are currently available. In addition, we conducted an additional experiment to compare the memory required by each method in addition to run-time. We will include this additional ablation study in the paper. Note that ( ) represents inference time for aggregator and the results are shown below:
>
> ||Aggregation|Memory [GB]|Run-time [ms]|
> |------|---|:---:|:---:|
> |NC-Net|4D Conv.|1.2|193.3(166.1)|
> |SCOT|OT-RHM|4.6|146.5(81.6)|
> |DHPF|RHM|1.6|57.7(29.5)|
> |CHM|6D Conv.|1.6|47.2(38.3)|
> |CATs|Transformer|1.9|34.5(7.4)|
>
> Given the results, we found that compared to other cost aggregation methods, CATs shows comparable efficiency in terms of computational cost. Note that NC-Net utilizes a single feature map while CATs utilize multi-level feature maps. Also, given similar memory consumption, our work remains beyond comparison in terms of run-time. For a future work, further improvements to CATs could be made by leveraging some of the recently proposed efficient attention modules (e.g.,[b]).
>
>     3. The authors have not discussed the societal impact of this work.
>
> CATs can be beneficial in a wide range of applications including semantic segmentation, object detection, action recognition and image editing.  For example, some methods for semantic segmentation tasks require cost volume aggregation. Such adoption would enhance the performance, which could affect various applications, e.g., autonomous driving.  On the other hand, our module risks being used for malicious works, which includes image surveillance system, but on its own, we doubt that it can be used for such works (L40, Supplementary). Our method may also be used to make deep fakes as Reviewer FB1A pointed out. We will include this more detailed discussion in the supplementary material.
>
> [a] Vaswani et al., “Attention is All you need”, NIPS’17
>
> [b]  Katharopoulos et al., “Transformers are RNNs: Fast autoregressive transformers with linear attention”, ICML’20

---

### Author Response · Authors · 2021-08-09
**General response**

We thank all the reviewers for their feedback, and are happy that our work was found to be “sound, intuitive and novel”, “bring significant improvements”, “could be adopted in other domains”, and “the flow of the paper is easy to understand”.
As we received reviews from almost all the reviewers suggesting to improve the quality of the presentation throughout the paper, we will definitely work on this to make the maximized improvements. Specifically, we will provide a high-level explanation before using any jargon and provide sufficient background knowledge, which will help to understand our paper better.

In the rebuttal, we provided additional experiments to compare the computational cost among existing methods, included detailed societal impact, explained the reasoning behind decomposing *inter*- and *intra*- operations, and added some of the missing results of CHM. Moreover, by conducting additional ablative experiments, we justified that the use of transformers and cost volume was essential, included the missing reference of COTR, proposed a method to mitigate the issue of order invariance, and discussed limitations of CATs.

---

### Author Response · Authors · 2021-09-16
**Final response**

We are glad that the reviewers found that our rebuttal provided reasonings with sufficient backups, which helped to resolve the main concerns raised by reviewer FB1A and CGvU. We also agree that the justification to design of our proposed method (why transformer and cost map) is indeed extremely important that with the discussion we made in the rebuttal, our paper would be strong as ever. We highly appreciate both reviewers for this.

However, reviewer FB1A leans toward rejection as incorporating such changes (experiments and analysis we made for rebuttal) requires a significant amount to rewrite, which we believe that it does not require much effort, but rather it is quite a light workload. Results, analysis and discussion were already made during the rebuttal, and we can just simply include them in the paper without considering additional experiments or writings. In light of this and as we empirically justified all the choices we made, which successfully addressed the suggestions and concerns the reviewers made, we kindly ask the reviewer to reconsider the rating.

Thanks

---

### Decision · Program_Chairs · 2021-09-27

**Decision:**

Accept (Poster)

**Comment:**

This paper initially had mixed reviews (7,7,6,3). The reviewers seem to agree that the proposed method is interesting, new enough to warrant publications, and the experiments are solid. The negative reviewer raised concerns about the paper's presentation, including lack of justification of certain technical choices and statements. Thanks to the author's rebuttal, many of these concerned have been addressed (also thanks to the additional experiments). The negative reviewer upgraded their rating to 5, saying they are convinced by the new experiments that the reviewers have added. In a private discussion between the reviewers (not visible to the authors), an approximate consensus emerge: the amount of revision required to improve the presentation is considered small enough to warrant acceptance now. The reviewers explicitly said that they trust the authors to make the promised changes.